

# Seasonally frozen soil modifies patterns of boreal peatland wildfire vulnerability

Simon J Dixon[1*]; Max C Lukenbach[2]; Nicholas Kettridge[1]; Kevin J Devito[3]; Richard M Petrone[4]; Carl A Mendoza[2]; J Michael Waddington[5]

[1] School of Geography, Earth and Environmental Sciences, University of Birmingham, Edgbaston, Birmingham, B15 2TT, UK.

[2] Department of Earth and Atmospheric Science, University of Alberta, Edmonton, AB, T6G 2E3, Canada

[3] Department of Biological Sciences, University of Alberta, Edmonton, AB, T6G 2E9, Canada

[4] Department of Geography and Environmental Management, University of Waterloo, Waterloo, ON, N2L 3G1, Canada

[5] School of Geography and Earth Sciences, McMaster University, Hamilton, ON, L8S 4K1, Canada

* - Corresponding Author: s.j.dixon@bham.ac.uk



## 1 Abstract

Peatlands play a vital role in the global carbon cycle, acting as one of the most important
global carbon sinks. However, an understanding of their environmental processes,
particularly in relation to a changing climate, remains inchoate. In particular, the role
seasonal ice or frost layers play in altering spring water balance, and thus vulnerability to
deep smoldering combustion during wildfire is not fully understood. Continental boreal
peatlands are characterized by periodic wildfire disturbance, which releases carbon, but can
also inhibit short-term peat productivity and carbon sequestration as the peatland recovers,
with recovery timescales linked to the severity or depth of burning. The presence of seasonal
frost layers coincides with drier spring conditions and an enhanced risk of wildfire. Two-
dimensional numerical modelling was conducted using HYDRUS-2D, a variably saturated
flow model, to simulate water balance in the vadose zone and assess vulnerability to fire
during prolonged rain free periods in the presence of continuous and discontinuous frost.
Our results show there is a lack of horizontal water transfer which increases spatial
variability in water balance and leads to pronounced heterogeneity in the risk of smoldering
combustion and the potential for deep combustion at hummock-hollow interfaces. Peatlands
are broadly divided into areas which are characterized by a dry near-surface and high water
contents at depth (water conserving), and those with a wetter near-surface, but
comparatively lower water contents at depth (productive). Those areas with dry near-
surfaces will be more vulnerable to wildfire and characterize around 50% of hummocks and
25% of hollows. In the presence of a seasonal frost layer productive peat layers in hollows
will show substantial drying out due to the frost layer disconnecting the surface from the
water table; this approximately doubles the proportion of hollows vulnerable to wildfire.
Breaks in the frost layer allows areas to maintain hydrological connectivity to a falling water
table, but this connectivity is limited in lateral extent and can drive further spatial
heterogeneity in vulnerability to wildfire ignition in the weeks when the frost layer begins to
thaw.





## 1. Introduction

Peatlands are an important global carbon sink (Frolking and Roulet, 2007;Gorham, 1991), accounting for up to 30% of the total soil carbon pool despite covering 3% of the global land surface (Gorham, 1991;Smith et al., 2004;Yu et al., 2010). Wildfire is the largest natural disturbance impacting northern peatlands (Stocks et al., 2002) and is increasing in areal extent (Turetsky et al., 2002). From 1960 to 1990, the annual area burnt across boreal North America doubled (Kasischke and Turetsky, 2006), and is projected to increase 118% by 2100 (Flannigan et al., 2005). Peatland carbon stocks are generally resilient to wildfire, returning to a net carbon sink approximately 13 years post-fire (Wieder et al., 2009). However, there are concerns that increased drying under future climates (Roulet et al., 1992) will increase wildfire severity (Turetsky et al., 2011), potentially transforming peatlands into a net carbon source (Turetsky et al., 2004) and leading to the long term degradation of their carbon stocks (Kettridge et al., 2015b).

Smouldering is the dominant form of combustion of peatland carbon stocks (Benscoter et al., 2011), with burn depths typically ranging from 0.05 – 0.10 m (Benscoter and Wieder, 2003;Lukenbach et al., 2015b;Shetler et al., 2008;Hokanson et al., 2016). Burn severity and smouldering (measured as depth of burn, DOB) are strongly controlled by the gravimetric water content (GWC) through the peat profile (Prat-Guitart et al., 2016;Benscoter et al., 2011;Lukenbach et al., 2015b). The GWC of peat is a function of water content and peat density, GWC thus determines the balance between a continued energy source to sustain combustion, produced by the burning of peat, and the energy sink (Benscoter et al., 2011;Prat-Guitart et al., 2016). Where GWC is low, small amounts of energy from the combustion of overlying or adjacent peat are sufficient to drive off water held within the peat to propagate smouldering (Benscoter et al., 2011). Therefore, patterns of peat moisture content and bulk density, which control peat GWC, are crucial to determining patterns of smouldering during peatland wildfire.

Complex interactions between peat hydrophysical properties (*e.g.* bulk density, unsaturated hydraulic conductivity) and atmospheric water demand control the distribution of peat moisture contents in the vadose zone, thereby influencing GWC and peat burn severity. However, boreal and sub-arctic peatlands are also characterised by pronounced seasonality.





During winter, snow typically covers peat soils in many places that are either completely frozen, or have frost lenses below the surface. Herein, we refer to this seasonally frozen soil as a "frost layer". Following spring snowmelt, frost layers thaw unevenly producing a heterogeneous landscape composed of frost free and frozen profiles (Petrone et al., 2008). In parts of Alberta this prolonged period of frost thawing can last into the summer months due to the insulating effect of the near-surface peat, potentially decoupling peatland vadose zone processes from the saturated zone (Thompson and Waddington, 2013). Decoupling may prevent water lost through evapotranspiration being replenished by limiting the upward capillary flow from the water table, inducing substantial drying of the near-surface. This enhanced drying would coincide with a spring period of heightened wildfire risk in continental boreal peatlands (Stocks et al., 2002) and may enhance early season wildfire burn severity (Turetsky et al., 2011).

Seasonal ice dynamics in peatlands not only have the potential to impact carbon loss during combustion, but may also influence the ecological trajectory post disturbance. Wildfire burn severity directly impacts peatland post-fire ecological recovery (Lukenbach et al., 2016), impacting the ability of the peat forming mosses to re-establish. Specifically, the interaction between pre-fire species cover and burn severity have a large influence on post-fire water availability (Lukenbach et al., 2015a, 2016), thereby playing a large role in the recolonization of peat-forming species. Therefore, if seasonal frost layers influence peat burn severity by altering GWCs at the time of wildfire it could change short-term carbon cycling in these landscapes.

In this study, we characterise near-surface frozen soil thaw dynamics within an unburned boreal peatland at a high spatial resolution. We simulate the control of this frozen layer on near-surface peat moisture dynamics using HYDRUS 2D, with specific emphasis on the ability of frozen layer to disconnect saturated water stores from the near surface. We assess how this control is modified by gaps within the frost layer, and consider the associated impact on early season wildfire severity and post-fire peatland recovery. The study has three objectives, to; i) use field measurements to characterise changes in the spatial extent of frozen soil at high spatial resolution within a northern peatland from snowmelt until the disappearance of frost, ii) explore how hydraulic properties and frost layer continuity



interact to drive vertical and lateral transfers of water during prolonged rain free periods of
high fire risk, iii) explore how spatial variability in hydraulic properties, peatland
microforms and frost layers interact to induce spatial variability in GWC and associated
smouldering severity during the exceptionally dry periods which precede wildfires.

## 2.Methods

### 2.1 Study site

Measurements were undertaken within a peatland (55.8° N, 115.1° W) within the lacustrine
clay region of the Utikuma Lake Research Study Area (URSA) in the Boreal Plains ecotone
(Devito et al., 2012), which has not burned since ~1935.  It is characterized by hummock
hollow microtopography, with ground layer vegetation consisting of *S. fuscum, S.*
*angustifolium* and *feather moss*. Vascular vegetation cover includes *Ledum groenlandicum,*
*Rubus chamaemorus, Maianthemum trifolia, Vaccinium oxycoccus* and *V. vitus-idea,* and the
canopy is comprised of black spruce (*Picea mariana*) with a basal area of 11 m² ha⁻¹ and an
average height of 2.3 m. For a full site description see Thompson and Waddington (2013)  .

### 2.2 Field Data

To examine spatiotemporal variations in depth to ice (objective *i*), depth to ice was measured
bi-weekly along a 50 m transect every 0.25 m (to a maximum depth of 0.5 m) through the
initial stages of the growing season (following snow melt) until no ice was detected (10th
May to 22nd July). Depth to ice was measured by pushing a metal rod through the peat
profile until solid frozen soil was detected. Near-surface peat moisture content was also
measured with a Delta-T theta probe at each point on the transect. Each sampling location
was also classified as a hummock or hollow, and the species cover was noted. Further the
relative elevation of each position along the transect was determined using a hose level
gauge.

### 2.3 Numerical Modelling

#### *2.3.1 Model Description*

Simulations were conducted using Hydrus-2D (Šimůnek et al., 1999), a two-dimensional
finite element model for simulating water flow in a variably saturated and unsaturated
medium with the model domain, discretised as a triangular grid. The governing flow
equation is a modified version of the Richard's equation:






$$\frac{\partial \theta}{\partial t} = \frac{\partial}{\partial x_i}\left[K\left(K_{ij}^A \frac{\partial h}{\partial x_j} + K_{ij}^A\right)\right] - S,$$  1


where $\theta$ is water content, $K$ is hydraulic conductivity, $h$ is pressure head and $S$ is the sink
term. Water retention is characterised by the Van Genuchten (1980)    model:

$$\theta(h) = \left\{\theta_r + \frac{\theta_s - \theta_r}{(1+(a|h|)^n)^m \theta_s}\right\} \quad h < 0 \; h \geq 0,$$  2


and:

$$m = 1 - \frac{1}{n} \quad n > 1,$$  3


where $\theta(h)$ is soil water retention as a function of the pressure head $h$, $\theta_r$ and $\theta_s$ are the
residual water content and saturated water content for the media respectively, $\alpha$ is an
empirical parameter related to the inverse of air entry pressure (m$^{-1}$) and $n$ is an empirical
parameter for the pore size distribution. Unsaturated hydraulic conductivity ($K$) is a
function of saturated hydraulic conductivity ($K_s$) and pressure head:

$$K(h) = K_s S_e^L \left(1 - \left(1 - S_e^{1/m}\right)^m\right)^2 \quad h < 0$$  4

$$K(h) = K_s, \; when \; h \geq 0$$

$$S_e = \frac{\theta - \theta_r}{\theta_s - \theta_r}$$


$S_e$ is the effective saturation and $L$ is a dimensionless pore tortuosity parameter (Simunek et
al., 1998).

### 2.3.2 Modelling domain

Two modelling domains were constructed to address objectives *ii* and *iii* (regular and
microtopography domains). The model domain for objective *ii* comprised a two-dimensional
grid in the xz plane, 5 m wide and 2 m thick, with a grid discretisation of 3 mm and a
horizontal stretching factor of 5. During numerical simulations, it was assumed that no flux
occurred across frozen layers based on the very low hydraulic conductivity of frozen soil
(McCauley et al., 2002). This was implemented to simulate a frozen layer with zero water
flux, inactive cells were included within this basic model framework; these rectangular



geometric objects were situated at a depth of 0.15 m below the soil surface. The inactive objects are 0.10 m thickness with a width specified by the modelling scenario. Grid refinements are applied to points at the evaporating surface and point on the upper edge of the geometry representing the frost layer; this refinement yields a finer model grid nearer the atmospheric boundary and around the no flow (frozen) layer, compared to the base of the grid. Moisture contents were recorded at the evaporating surface, and depths of 0.05, 0.10, 0.15 and 0.50 m.

The model domain for objective *iii* simulated a peatland microtopography sequence of hummocks and hollows. The model domain was a grid in the xz plane, 5 m wide. The hummock-hollow surface topography was characterised as a continuous curve from the top of the hummocks to the base of the hollows. The hummock-hollow sequence had an amplitude of 0.4 m and wavelength of 2 m. The model domain contains a total of 2.5 hummock-hollow sequences. Therefore, the depth from the hummock and hollow surface to the base of the model domain was 1.4m and 1.0 m, respectively.

### 2.3.3 Boundary and initial conditions
The initial conditions of the regular model domain were set assuming an equilibrium pressure head through the peat profile. For the planar surface model domain, the starting water table (zero pressure head) was set at a depth of 0.05 m below the evaporating surface. In the case of the microtopography domain, the water table was set to a depth of 0.05 m below the lowest point of the hollow. The base and sides of all model domains were set as no flow boundaries and the surface as an atmospheric boundary. A time variable flux was applied to the atmospheric boundary, representing the diurnal variation in evapotranspiration over a 50 day modelling period, assuming no rainfall input (Dixon et al., 2017). We therefore aim to model a prolonged period of potential drying which would typically precede a spring wildfire. For example, within the fire prone regions of the western boreal plain, Canada, between 1922 and 2007, 10% of the months of May had less than 16 mm of precipitation, with a minimum of 5.6 mm (Slave Lake, Environment Canada, 2017). A threshold value for when surface tension inhibits evaporation (hCritA) is employed. Here we apply a value of 400 mb to represent vegetative stress within mosses limiting evaporation (McCarter and Price, 2014). The frost layer is modelled as rectangular blocks in the model domain with all edges set as no flow boundaries; we thus model the frozen layer



as a static part of the domain without freeze-thaw mechanics. We are therefore modelling
the effect of the frost layer only on disconnecting the near-surface of the model domain from
the deeper saturated layer. This is an exploratory modelling framework to examine the
impact of frost layer disconnection that does not account for the recharge effects of water
provided from thawing frost during the model run. The magnitude of this potential daily
recharge rate will be determined from observed rates of ice melt and considered in the
context of wider modelling assumptions, notably evaporation rate, initial moisture
conditions, and hCritA. The modelling results should therefore be seen not in terms of
absolute predictions, but rather comparisons in the response, and magnitude of response,
between different scenarios.
The microtopography model was 'spun-up' for a period of 200 days to generate realistic
starting moisture conditions as using a default of equilibrium pressure heads in HYDRUS
generates unrealistically dry hummocks, whereas naturally hummocks are characterised by
relatively moist conditions a few centimetres below the surface (Benscoter and Wieder,
2003;Thompson and Waddington, 2013). The spin up period was applying a weekly rainfall
recharge to the peat surface equal to moisture lost through evapotranspiration. This spin up
achieved a dynamic equilibrium in water contents with a planar water table depth of
approximately 0.04 m from the base of the hollows and higher unsaturated moisture
contents in the hummocks (See supplemental material).
*2.3.4 Hydraulic Properties*
Peat hydraulic properties vary across several orders of magnitude (e.g. Hogan et al.,
2006;Lewis et al., 2012;Kennedy and Price, 2005;Boelter, 1965;Branham and Strack,
2014;Beckwith et al., 2003;Baird et al., 2008;Baird et al., 2016). These hydraulic properties
have a crucial control on the peatland response to evaporation (Kettridge et al., 2015a;Dixon
et al., 2017). However, using mean values of these properties in modelling investigations
fails to accurately characterise either the average response or a typical range of responses for
a given distribution of peat properties (Kettridge et al., 2015a;Dixon et al., 2017). Therefore,
we apply a distribution of hydraulic properties reported in *Dixon et al* [2017], based upon
field data collected by *Lukenbach et al* [2015] and *Thompson and Waddington* [2008] (Table I).





These data encompass a full range of peat types and properties, from centre of bogs to dense
margin peat areas.
*Kettridge et al* [2015] found that the key factors controlling whether a peat profile displayed
high surface tensions under evaporation (*water conserving*) or was able to maintain low
surface tensions during evaporative stress (*productive*) were inverse entry of air pressure ($\alpha$)
and saturated hydraulic conductivity ($K_s$); where higher $\alpha$ and lower $K_s$ correspond to a
greater likelihood of peat being water conserving under stress. In this study, due to
computational constraints, we conceptually define a vector through hydraulic property
space along the axis corresponding to $\alpha$ and $K_s$. We define three different combinations of $\alpha$
and $K_s$ along this vector to represent profiles across the transition from water conserving to
productive, and generate these values for: all peat, just hollows and just hummocks.  Values
for $\alpha$ and $K_s$ applied were the mean, and plus and minus one standard deviation from the
mean. Peat hydraulic properties were not varied with depth in the model. Although
variations in peat hydraulic properties with depth have been shown (e.g. Sherwood et al.,
2013), *Kettridge et al.* [2015] found only weak dependence on depth for values of $\alpha$ and $K_s$.
*Quinton et al* [2008]    also showed that $K_s$ is dependent on the degree of compaction and
decomposition, which does not necessarily show a linear relationship with depth.
A combination of low $K_s$ and high $\alpha$ inhibit water flow and tend towards high surface
tension under evaporative stress; defined here as "*water conserving*". Conversely, high $K_s$ and
low $\alpha$ readily transport water to the evaporating surface and is defined as "*productive*"
(Table I). We keep values of residual water content ($\theta_r$), saturated water content ($\theta_s$), pore
tortuosity ($l$) and $n$ constant. For modelling scenarios with a hummock-hollow sequence we
further generate values for $\theta_s$, $l$ and n, as well as values for $\alpha$ and $K_s$ (plus and minus one
standard deviation) for hummocks alone (Table I). It is important to note that the difference
in modelled hydraulic properties between hummocks and hollows is not as great as the
difference between the peats classified as water conserving or productive. The hydraulic
property type, based on the property distributions, is more important to behaviour than
whether the peat is a hummock or hollow. Hydraulic properties have been shown to have a
relationship to moss species (McCarter and Price, 2014), however this remains poorly





understood   and   more   detailed   field   data   collection   is   needed   to   parameterise   these
relationships in numerical models.

| Material | $\theta_r$ | $\theta_s$ | $\alpha$ | $n$ | $K_s$ (cm/hr) | $l$ |
|---|---|---|---|---|---|---|
| Mean | 0.01 | 0.939 | 1.828 | 1.192 | 18.31 | -1.411 |
| Water Conserving | 0.01 | 0.939 | 2.380 | 1.192 | 16.31 | -1.411 |
| Productive | 0.01 | 0.939 | 0.176 | 1.192 | 20.31 | -1.411 |
| WC Hummock | 0.01 | 0.965 | 14.999 | 1.213 | 16.31 | -1.411 |
| PR Hummock | 0.01 | 0.965 | 0.628 | 1.213 | 20.31 | -1.411 |

Table I – hydraulic properties of peat used in the modelling scenarios and based on data in Dixon et
al (2017)

### 2.3.5 Modelling Design

To examine water flow pathways and the water balance in the presence of continuous and
discontinuous frost layers (Objective *ii*), four different frozen soil geometries were created
within the regular model domains: *i*) a continuous layer of frozen soil/frost across the model
width; *ii*) a 0.5 m wide gap in the centre of the frost layer; *iii*) two 0.5 m wide gaps in the
frost layer either side of a central 1 m wide block of frozen soil; *iv*) frost free. Mean, water
conserving and productive peat hydraulic properties were used to parameterise the peat
layers in the four model geometries, giving a total of 12 model scenarios. For Objective *iii* the
hummock-hollow   sequence   within   the   microtopography   modelling   domain   was
parameterised with different hydraulic properties in the different microtopographical units.
Moving from left to right the model domain was parameterised as; a water conserving
hummock, a water conserving hollow, a productive hummock, a productive hollow, and a
water conserving hummock. The sequence ensures all four possible transitions between
hummock and hollow properties are represented in the model domain. The remainder of the
model domain was designated as having mean peat hydraulic properties.

### 2.3.6 Model Analysis

To assess fire severity, simulated volumetric moisture contents (VMC) were converted to
gravimetric water contents:

$$GWC = \frac{\theta}{\rho} \qquad\qquad 5$$


where *GWC* is gravimetric water content and $\rho$ is the mean density of peat for a given
sample.





A probabilistic approach is taken to estimate the likelihood that the gravimetric water
content of given sub-set of peat (Table I) at a given time in a model simulation is lower than
a threshold for smouldering wildfires ignition. A normal distribution of gravimetric water
contents was calculated, multiplying the VMC by the full normal distribution of peat
observed peat densities, given by:

$$Y \sim N\left(\theta\mu_\rho, \theta^2\sigma_\rho\right) \qquad\qquad 6$$


where Y is a normal distribution with a mean of $\theta\mu_\rho$ and a standard deviation of $\theta^2\sigma_\rho$. For a
given gravimetric water content, in this case corresponding to the threshold for smouldering
(GWC=250%), the z score for a given distribution is computed as:

$$z = \frac{x - \theta\mu_\rho}{\theta^2\sigma_\rho} \qquad\qquad 7$$


where $x$ is the smouldering threshold. A probability of the gravimetric water content being
lower than the threshold for a given point and at a given time is expressed as a cumulative
normal distribution function:

$$P = \frac{\theta^2}{\sigma}\frac{1}{\sqrt{2\pi}}\int_{-\infty}^{x} e^{\frac{(t-(\mu/\theta))^2}{2(\sigma/\theta^2)^2}}\,d \qquad\qquad 8$$


The results from Eq. (8) can then be plotted over time as a metric to indicate the probability
of deep smouldering as a function of gravimetric water content probability, given the
distribution of peat density in a given type of peat.
## 3. Results
### *3.1 Ice field measurements*
Shortly after snowmelt, the frost layer largely followed the surface topography at a depth of
0.2-0.3 m and subsequently retreated slowly as the frozen soil nearest to the surface began to
thaw (Figure 1). The frozen layer eventually began to break up and disappear completely
over a period of five weeks from 22nd June, when the frost layer is patchy and broken though
in places, to 1st August, when the transect was frost free. The recession of the frozen soil
layer can be taken as a daily average from 10th May to the start of break up on 22nd June and
along with specific yield of average peat properties from our data set, gives an average daily



moisture recharge rate of 1.8 mm/day. During the period of frost break up, sections of frozen
soil 1-1.5 m in length persist whilst the frost in other areas has thawed completely. Although
these persistent blocks of frozen soil can be associated with either hummock or hollow
microtopography, the frost layer tends to begin to break up earlier in hollows than
hummocks. Furthermore, some of the more persistent blocks of frozen soil are in the bases of
hummocks. A binomial logistic regression was run on the effects of topography of the
presence of frost on 13th July. The Hosmer-Lemeshow test shows the model fits the data well
(p=446), with topography predicting presence of frost at a significance level of $\alpha$=0.90
(p=0.091).

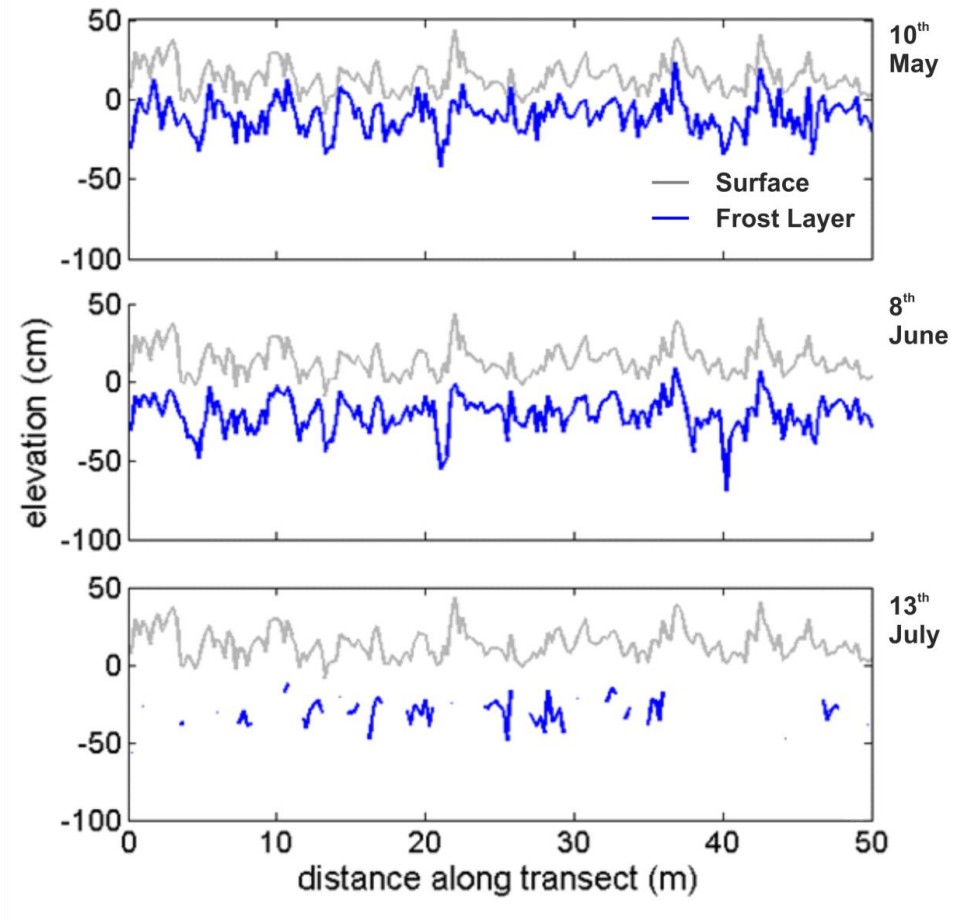


Figure 1 – Field measurements showing surface topography and initial depth to
ice (i.e. top of the frost layer) and the breakup of ice into a discontinuous layer
later in the season (note the vertical exaggeration in scale).





*3.2 2-D model simulations (topography excluded)*
Trends in water content and water movement were strongly associated with whether peat
hydraulic properties were "water conserving" or "productive" as well as the continuity of
the frost layer. For scenarios with solid frost layers (Figure 2), water is initially lost from the
near-surface of the "water conserving" peat via evaporation (Figure 2); within the near
surface the volumetric water content (VWC, θ) declines to ~0.26 after three days (Figure 2a)
and quickly raises near-surface tensions limiting evaporation. As a result, after a week of
evaporation, VMC in the top 0.05 m of peat profile is low (Figures 2a and 2b). However, the
water table did not drop below 0.10 m until day 26 (VWC equals the saturated water content
of $\theta_s = 0.94$; Figure 2c), with a saturated zone remaining above the frozen layer (Figure 2d; $\theta =$
$\theta_s$). In comparison, for "productive" combinations of peat hydraulic properties, surface
tensions did not rise to levels that limit evaporation. Profiles thus continued to evaporate
and the water table dropped to the depth of the frozen layer on day seven of the simulation

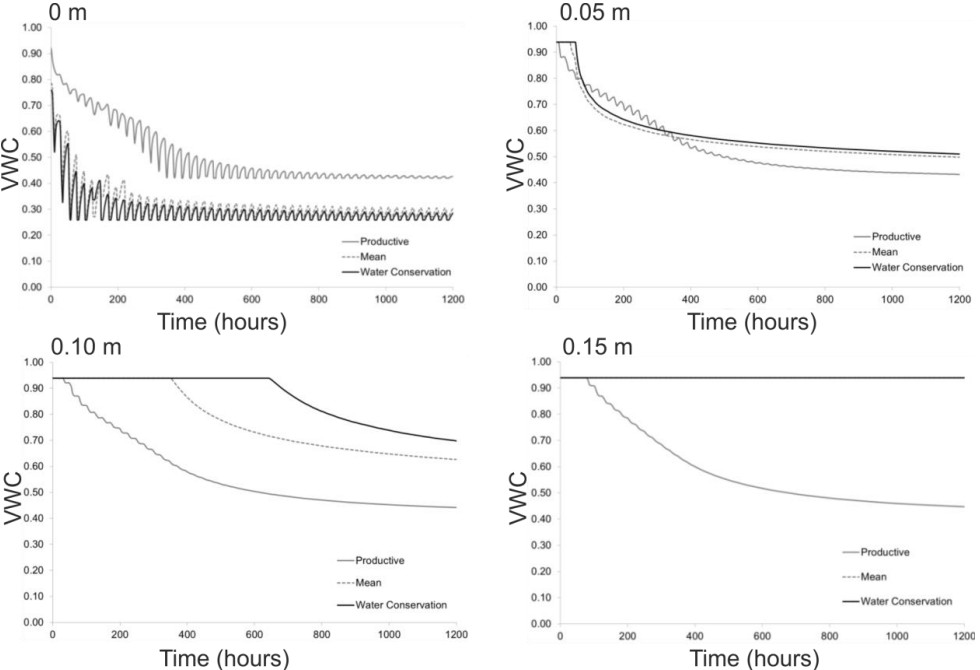


Figure 2 –Volumetric Water Content over time for solid frozen soil layer at 0.20 m
depth. Different panels show VWC at different depths in the peat profile for three
different set of peat hydraulic properties; productive, mean and water conserving.





(Figure 2d; VWC declined after day 7). Thereafter, the VWC in the peat layer above the frost
layer continues to decline, as evaporation is sourced from unsaturated zone storage above
the frost layer. After three weeks of evaporation, the VMC at 0.15 m depth is less than 0.50
(Figure 2d).
For scenarios with one or more holes in the frost layer, simulated VMC for the mean and
water conservation hydraulic properties are the same as for the solid frost layer. This is
because the water table does not drop to the depth of the frost layer during simulations.
Therefore, continuity in the frost layer is only important for "productive peat" scenarios
(Figure 3). For productive peat, holes in the frost layer result in spatiotemporal variations in
water contents and tensions as water is transported to the surface from the saturated peat
below the discontinuous frost layer (Figure 3b/c).  Once the water table drops below the
level of the frost layer, the near-surface peat over a hole in the frost layer is able to maintain
a VWC of $\theta \approx 0.50$. However, within the vadose zone, lateral transfer of water supplied
through the gap is very limited (Figure 3bc). After five weeks of evaporation there is a clear
difference in near-surface VWC between the frozen soil scenarios (Figure 3). The frost free
(Figure 3a) and solid frost (Figure 3d) scenarios represent the two extremes in VWC, with
the discontinuous frost layer scenarios (Figures 3b/c) showing characteristics of both solid
frost and frost free scenarios. Where there is a break in the frozen layer, VWC immediately
above the break corresponds to the frost free scenario (Figure 3a). However, a short lateral
distance away from the break the VWC in the discontinuous frost scenarios closely
resembles the solid frost scenario (Figure 3d).
Once the water table has reached the depth of the frost layer the near-surface peat above the
layer does not replace water lost through evaporation and thus peat VWC continues to
decline until near-surface water tensions reach 400 mb and evaporation is limited.
Conversely, above the breaks in the frost layer the near-surface water tensions remain in the
range 150-350 mb over the whole diurnal cycle, indicating that water supplied from deeper
saturated peat is able to maintain some evaporation. Consequently, discontinuous frost
layers generate heterogeneity in near-surface VWC; $\theta \approx 0.60$ above breaks in the frozen layer
compared to $\theta \approx 0.45$ above the frozen layer (Figures 3b and 3c).



Figure 3 – Water balance in four different frost layer model scenarios after five
weeks of diurnal evaporation. This shows breaks in the frost layer allow the
evaporating surface to maintain connectivity to the falling water table below the
ice, but there is limited lateral connectivity with little water supplied to areas not
directly above the hole.




### 3.3 2-D model simulations (topography included)

Hummock-hollow microtopography simulations show lateral water transfer between
adjacent water conserving and productive zones that substantially influences GWC and
associated wildfire severity. Although the spatial extent of lateral water transfer is limited,
and its influence doesn't extend beyond 0.5 m, this can induce strong spatial variability in

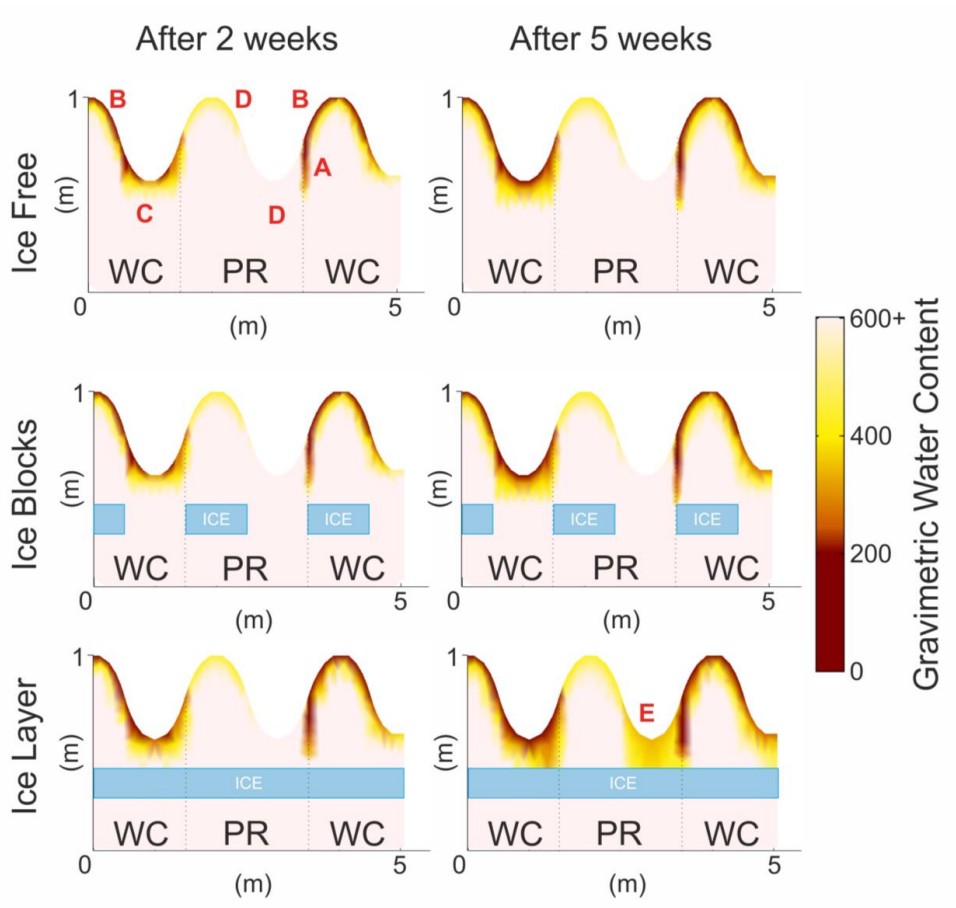

Figure 4 – Gravimetric water contents across modelled domains after two and five
weeks of evaporation. WC = Water conserving peat type, PR = Productive peat
type. The water table intersects the frost layer on day 26, as a result there are
only minimal differences in water content distributions between the different ice
scenarios after two weeks. For the results after five weeks, where the water table
is below the level of the frost layer, the surface moisture content is partly
determined by connectivity to the falling water table. This shows productive areas
above solid ice will dry out substantially compared to the same areas over
discontinuous or absent frost layers (See text for discussion of labels A-D).





GWC (Figure 4). Lateral water transfer results in zones of low GWC typically at the sides of peat hummocks (one example labelled as "A" in Figure 4). These represent zones of high susceptibility to deep smouldering combustion and thus high burn severity.

Evaporation and GWCs of a given profile through the transect depends both on the hydraulic properties of the profile itself, and the hydraulic properties of adjacent areas. After two weeks of evaporation, the water table has not declined to the depth of the frost layer, thus the ice and ice-free scenarios behave similarly (Figure 4a/b). Areas of water conserving peat, which respond to evaporative stress with high surface tensions, have very low near-surface GWCs.  As such, water conserving hummocks exhibit GWCs <250 % at depths of <0.03 m (e.g. "B" in Figure 4). Similarly hollows have GWCs <250% to a depth of 0.05m (e.g. "C" in Figure 4). At depth within both hummocks and hollows, deeper peat has much higher GWC. Conversely, areas with productive peat facilitate the upwards movement of water from deeper in the peat, resulting in a more even vertical distribution of GWC (e.g. "D" in Figure 4). This results in higher GWC at the near-surface of productive hummocks compared to water conserving hummocks. In productive hollows, this results in higher water contents at the near-surface and lower water contents at depth.

Five weeks into the simulation, GWC varies little between the frost free and broken frost layer scenarios (Figure 4); however, the scenario with a solid frost layer, shows that the GWCs of hollows are substantially lower compared to the other ice scenarios and the same ice scenario after three weeks (Figure 4). In productive hollows, after five weeks (labelled "E" in Figure 4), the surface GWC is 400%, compared 800% for the no ice scenario. In water conserving hollows surface water contents are similar for all ice scenarios.

Interactions between adjacent water conserving and productive areas create small regions of enhanced heterogeneity in GWC (labelled "A" in Figure 4). In the case of a water conserving hummock next to a productive hollow, water is drawn out of the side of the hummock and a region of very low GWC develops at the margin of the water conserving area. This region of low GWC is affected by the presence of a solid frost layer, with the area being slightly larger for the solid ice scenario compared to the other two scenarios after three weeks and substantially larger and also of much lower GWCs after five weeks (Figure 4).




*3.4 Differences between microtopographic position and hydraulic properties*

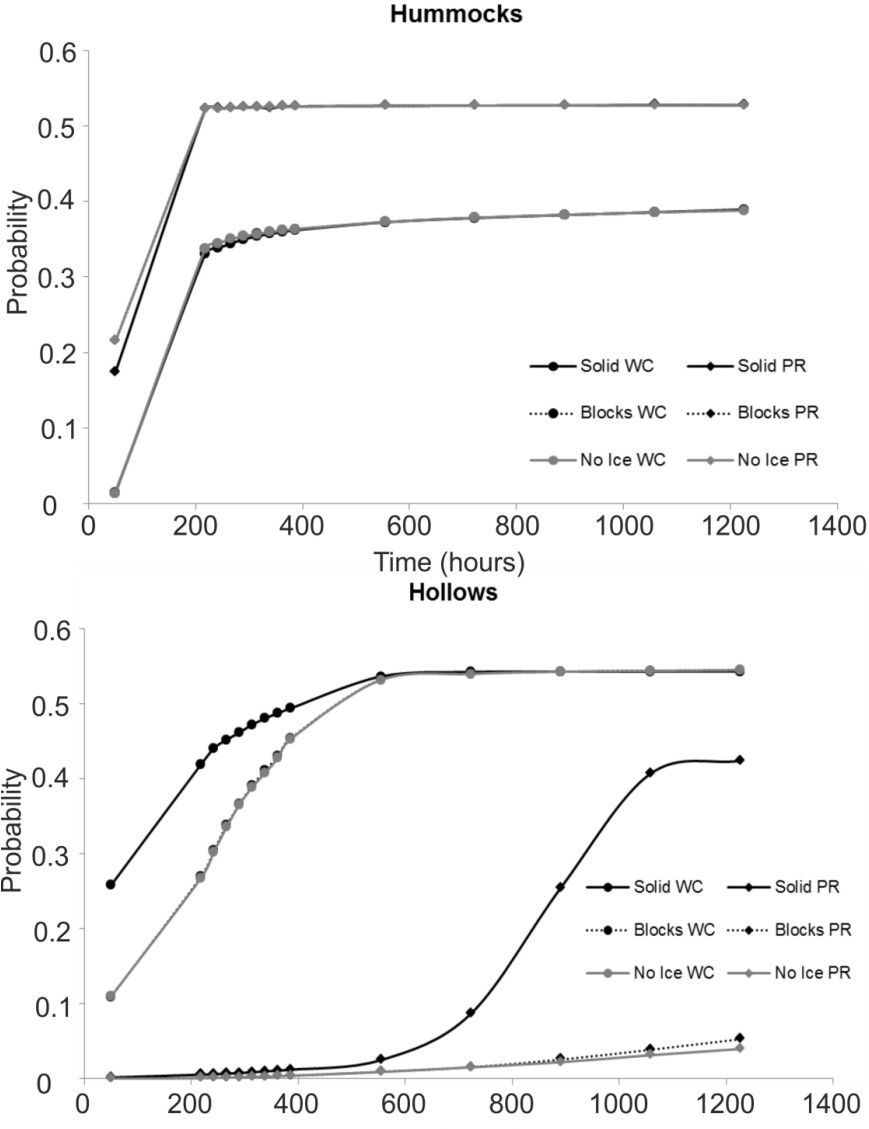

Figure 5 – The probability of ignition (gravimetric water content is lower than
250%), for near-surface peat, over time for the different microtopographic areas
and hydraulic properties (PR – Productive peat properties, WC – Water conserving
peat properties)
For hummocks, all scenarios show very little change in surface water content over the model
runs and show little difference between scenarios (Figure 4), a finding which is shown
quantitatively in Figure 5A. Probabilistic GWCs are calculated based on the normal

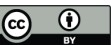



distribution of peat densities using Eq. (8). Results are displayed as the probability of GWC
being lower than 250%, representing a threshold for smouldering. All scenarios show similar
hummock probabilities over time for both hydraulic properties types (Figure 5A), indicating
the presence of a solid frost layer has only a small influence on hummock GWC. Conversely
solid ice has a substantial effect on hollow GWC (Figure 5B). In productive peat hollows
probabilities rise steeply two weeks into the simulation and the ice free and ice block

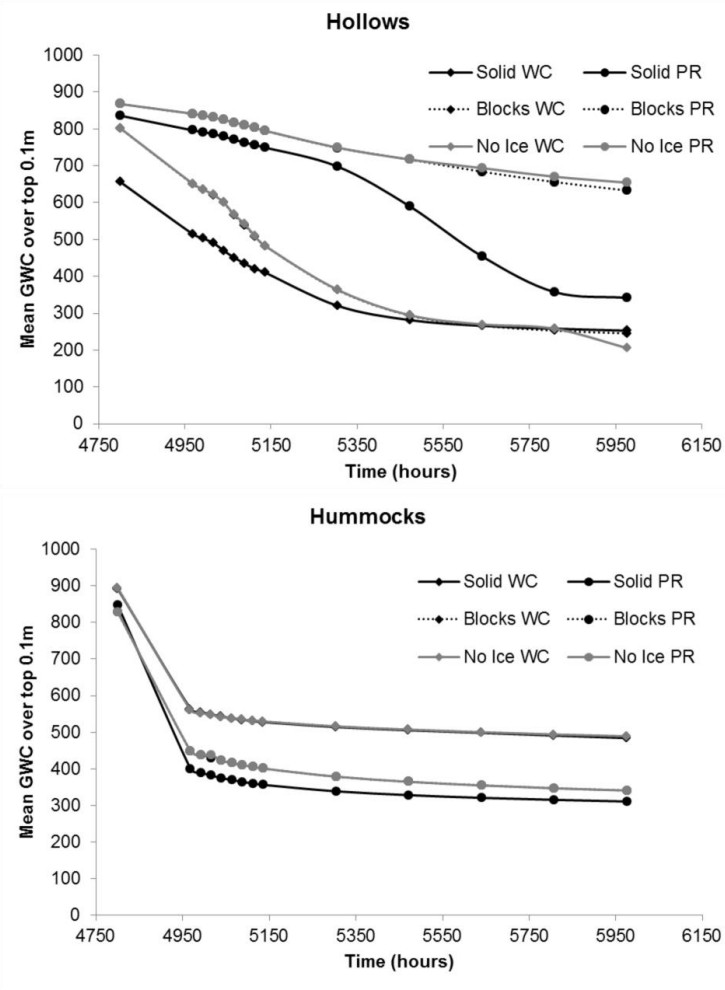


Figure 6 – Mean gravimetric water content over the top 0.1m of the profile for
different microtopographic features showing how vunerability to deeper
smouldering varies over time for different scenarios and features.
scenarios diverge (Figure 5B). Mean GWC over the top 0.10 m of the peat profile (Figure 6)
shows a similar pattern to Figure 5. Hummock GWC is relatively insensitive to ice scenario,



whereas for hollows the solid ice scenario shows generally lower GWC over the top 0.1m
compared to other scenarios (Figure 6A).

## 4. Discussion

Our results show that seasonal frost layers can affect near-surface GWC and thus the
vulnerability of peat to smouldering during wildfires. However, this increased vulnerability
probability is also dependent on peat hydrological properties and microtopographical
position.

### 4.1 Peatland vadose zone hydrology in presence of ice

In common with other studies, numerical modelling herein demonstrates that peat hydraulic
properties exert a primary control on near-surface water contents (Kettridge et al.,
2015a;McCarter and Price, 2014;Dixon et al., 2017). Notably, simulations indicate presence or
absence of a frost layer has a minimal effect on the water balance for peat hydraulic
properties which restrict the unsaturated water flow, and thus limit evaporation (*water
conserving*). In such case, a drier near-surface peat layer developed with volumetric water
contents of $\theta$=0.30-0.50 in the upper 0.05 m. However, below 0.10 m depth, peat remained
saturated over multiple weeks of evaporation (Figure 2). Therefore, even after six weeks of
evaporation, the water table remains at a depth no greater than 0.15m, which is higher than
the observed spring depth of ice in boreal peatlands (Figure 2, and Petrone et al., 2008).
Conversely, for peats that facilitate unsaturated water flow from depth to the evaporating
surface (*productive*), ice has a substantial effect on water contents. Although productive peat
readily supplies water to the evaporating surface and maintains a wetter near-surface
(Figure 2), greater evaporation, compared to water conserving peat, drove a rapid fall in the
water table and, subsequently lowered water contents at depth. In situations where
productive peat is underlain by a solid frost layer, the water table dropped to the level of the
ice, and subsequently the peat above of the frost layer continued to dry out with further
evaporation (Figure 3). Although surface tensions rose during drying in the layer above the
ice these did not become high enough to begin to limit evaporation until the peat at 0.15m
depth had a water content of $\theta$=0.60 and evaporation persisted even with $\theta$=0.45.

Horizontal gaps in the frost layer allow the evaporating surface to maintain hydrological
connectivity with the falling water table and thus avoid *productive* peat drying out (Figures 2



and 4). However, the primary unsaturated flow direction from depth is vertical and there is
limited lateral flow away from the frost gap. This results in heterogeneity of near-surface
water contents with areas directly above the frost gap maintaining near-surface water
contents similar to frost free scenarios, whereas more distant areas dry out to the same
extent as solid frost layer scenarios (Figures 2 and 3).

Where peat hydraulic properties are varied across a hummock-hollow microtopography
sequence water balance patterns become more complex compared to a planar atmospheric
boundary (Figure 3). As with a flat atmospheric boundary, *water conserving* hollows are
characterised by a very dry near-surface (upper 0.05 m), but with much higher water
contents at depths of greater than 0.1 m. Conversely, *productive* hollows are able to maintain
high near-surface volumetric water contents and maintain hydrological connectivity with a
falling water table. As with the planar surface scenarios the transfer of water from depth
through the frost gap is primarily vertical, with little lateral transfer of water away from the
gap. The combination of a discontinuous frost layer with variations in peat hydraulic
properties across a hummock-hollow microtopography leads to a high degree of
heterogeneity in volumetric and gravimetric water contents.

It is important to note however that our initial model conditions assume near saturation
following spring snow melt above the frost layer. The system is thus relatively wet, with a
shallow starting water table. A drier set of initial conditions, either as a result of different
hydrogeological conditions, a drier autumn period leaving the soil drier before winter, or a
smaller snowmelt recharge, would mean a lower starting water table and, therefore, an
increased probability that the water table will reach the frost layer, as observed by *Petrone et*
*al* [2008]. We also assume the frost layer does not thaw during the model runs and thus does
not provide water recharge, which is calculated as approximately 1.8 mm/day. We assume a
fixed daily evaporation rate of 4.5 mm/day; this is a fairly high rate, however, *Sphagnum*
evapotranspiration rates can reach an average of 0.47 mm/hr, and unlike systems which
show highest ET rates coincident with highest air temperatures, Alberta peatlands show
peak ET rates during the early growing season as frozen soil is thawing (Brown et al., 2010),
which is the period being modelled. Finally, the critical surface tension threshold (hCritA)





which shuts off evaporation, here a fixed value of 400mb after McCarter and Price (2014),
needs to be considered. It is reasonable to assume the true value of hCritA will vary with
relatively humidity and may in fact be higher at some points during the model runs which
represent prolonged rain free periods. As a result, the model scenarios may underestimate
total evaporation, but overestimate the drying given the lack of thawing frost recharge.
However, all these assumptions (no thaw, evaporation rate, hritA and initial conditions) are
within 1-2 mm/day, and are constant between scenarios.

Considering these assumptions, we restrict ourselves to exploring relative comparisons in
VWC and GWC between the modelling scenarios, rather than use modelling results to
deliver quantitative predictions. The overall effect of our modelling assumptions is likely to
mean that for a given real peatland the time taken to reach a given point in the results may
be under or overestimated, but the overall pattern of behaviour is likely to be consistent as
this is governed by the peat hydrological properties. If more water is lost through
evaporation (higher evaporation rate or dynamic HcritA), or the initial conditions are drier,
then the water table will recede to a deeper level earlier and the near-surface GWCs will fall
earlier. Conversely, if the recharge from thawing frost is accounted for under the same
evaporative and initial conditions the water table recession will be slower and it will take
longer for the near-surface GWCs to fall.

### 4.2 Implications for peatland wildfire burn severity

Our results suggest that seasonal frost has a pronounced effect on potential burn depths in
boreal peatlands during periods of pronounced drying. Regardless of the type of peat
properties, the frost table makes near-surface peat layers, particularly in hollows, more
vulnerable to wildfire (smouldering) following a multi-week period without rain by
reducing near-surface moisture contents. Given that burn severity in peatlands is highly
heterogeneous (Lukenbach et al., 2016), seasonal frost dynamics along with peat hydraulic
properties may help explain why some hummocks (or hollows) burn more severely than
others. Specifically, the uneven thawing of ice in spring may drive heterogeneity in
connectivity to the water table and facilitate variations in near-surface GWC. Pronounced
variability in burn severity within microform types has been observed (Lukenbach et al.,



2015a, 2016), and these results illustrate hydrological mechanisms beyond peat properties
which can drive variability in the probability of peat ignition during wildfire.

The probability that gravimetric water contents are lower than a critical smouldering
threshold of ~250% (c.f. Thompson et al., 2015;Zoltai et al., 1998;Lukenbach et al.,
2015b;Benscoter et al., 2011) are higher when a frost layer was present compared to when it
was absent, and was most pronounced for areas with productive peat hydraulic properties
(Figures 5). The most sensitive sub-set is productive hollows, where the presence of a frost
layer substantially increases the probability that near-surface GWC drops below 250%
(Figure 5) and also lowers the average GWC over the top 0.1m of the profile (Figure 6). Our
results indicate that ice free conditions coinciding with a prolonged rain free period in
peatland would result in approximately 50% of hummocks being vulnerable to smouldering
combustion (Figure 5A), with approximately 25% of hollows vulnerable to smouldering
combustion and potentially deep depths of burn (Figure 5B). In the presence of a solid frost
layer, however, the proportion of hollows vulnerable to appreciable smouldering depths
rises to around 50% (Figure 6B), effectively doubling the extent of deeper depth of burn.
Heightened heterogeneity in water content can occur when adjacent areas have different
hydraulic properties. A wedge of low GWC develops where either a water conserving
hummock is next to a productive hollow, or a productive hummock is adjacent to a water
conserving hollow (labelled as "A" in Figure 4). Results also indicate that although
hummocks are much higher above the water table, they are able to retain a moist vadose
zone (e.g. Benscoter and Wieder, 2003;Thompson and Waddington, 2013;Shetler et al.,
2008;Lukenbach et al., 2015b;Benscoter et al., 2011) and are to an extent hydrologically
disconnected from the water table with surface water balance dependant on water retained
within the hummock; this means they are characterised by a very dry near-surface, but a
moist enough interior of the hummock that GWCs offer protection against deep
smouldering, regardless of the presence or absence of a frost layer.

An interesting implication of these results is that productive hummocks may be better able
to maintain higher near-surface GWCs and thus resist smouldering combustion; this would
mean that they are better placed to recover quickly from wildfires (Lukenbach et al., 2015a,



2016). Conversely, more water conserving hummocks may be susceptible to higher burn
severity due to their very dry near-surface, and may then be unable to recover quickly post-
fire or continue to conserve water (Lukenbach et al., 2016, 2015a). The presence of seasonal
frost layers changes the implications of burn severity and recovery and introduces a trade
off in the optimum peat profiles. For hollows, a water conserving peat would be slower
growing and experience more frequent periods of water stress, however when ice is present
it will be characterised by a thin, dry layer above a relatively moist layer which may restrict
smouldering to a thin surface band of peat and promote more rapid recovery post-fire.
Conversely, hollows with productive peat will be able to buffer periods of water stress and
will be faster growing, however, they will be vulnerable to drying out in the presence of a
frost layer, meaning they could be subject to high burn severity and then take a long time to
subsequently recover ecohydrological function.

## 5. Conclusions

Two dimensional numerical modelling results show that seasonal frost layers can play an
important role in disconnecting the evaporating surface of the peatland from the deeper
water table and so can enhance heterogeneity in near-surface water contents, which likely
influences patterns of ignition and burn severity during wildfire. In common with other
studies, we find peat hydraulic properties provide an important control on peat water
balance, as they control the ability of the peat to transmit water from deeper saturated layers
to the evaporating surface. Hummock microforms are typically higher above the water table
and so over seasonal timescales do not depend on direct connection to the deep water table
to supply evaporative demand, rather they utilise water stored in a large unsaturated mass
of the hummock. As a result of their decreased dependence on connection to the water table,
hummock microforms are relatively unaffected by the presence of a seasonal frost layer.
Conversely, hollow microforms depend to a greater degree on maintaining connectivity to
the water table in order to supply water to meet evaporative demand. Hollows which tend
towards water conserving are less able to transmit water from deeper saturated layers will
tend to dry out at the very near-surface, but maintain high tensions and pressure gradients
to layers with higher water contents immediately below. In prolonged rain free periods,
regardless of seasonal ice presence, these areas are therefore vulnerable to deep smouldering
in wildfire. Conversely, hollows comprised of peat which tends towards greater production



and water use, are highly dependent on seasonal ice presence. Productive hollows will dry
out in the presence of a seasonal frost layer as their surface is disconnected from the water
table. In addition, they are able to maintain productivity, and evaporation, in the presence of
low water contents, and therefore can be subjected to substantial drying. This substantial
drying will make the area vulnerable to deep burning in wildfires. At the landscape scale
our results show the presence of a frost layer with a prolonged rain free period will increase
the proportion of hollows vulnerable to deeper burning (>0.1 m) from 25% to 50%. The high
degree of natural variability in peat hydraulic properties will lead to a high degree of
heterogeneity in near-surface water contents during rain free periods, and thus
heterogeneity in wildfire burning within microform type. The presence of a seasonal frost
layer both raises the overall risk of smouldering and deeper burning (>0.1 m) during
wildfires, but also enhances the degree of heterogeneity.



581 Baird, A. J., Eades, P. A., and Surridge, B. W. J.: The hydraulic structure of a raised bog

582 and its implications for ecohydrological modelling of bog development, Ecohydrology, 1,

583 289-298, 2008.

584 Baird, A. J., Milner, A. M., Blundell, A., Swindles, G. T., and Morris, P. J.: Microform-

585 scale variations in peatland permeability and their ecohydrological implications, Journal of

586 Ecology, 104, 531-544, 10.1111/1365-2745.12530, 2016.

587 Beckwith, C. W., Baird, A. J., and Heathwaite, A. L.: Anisotropy and depth‐related

588 heterogeneity of hydraulic conductivity in a bog peat. I: laboratory measurements,

589 Hydrological processes, 17, 89-101, 2003.

590 Benscoter, B. W., and Wieder, R. K.: Variability in organic matter lost by combustion in

591 a boreal bog during the 2001 Chisholm fire, Canadian Journal of Forest Research, 33, 2509-

592 2513, 10.1139/x03-162, 2003.

593 Benscoter, B. W., Thompson, D. K., Waddington, J. M., Flannigan, M. D., Wotton, B.

594 M., De Groot, W. J., and Turetsky, M. R.: Interactive effects of vegetation, soil moisture and

595 bulk density on depth of burning of thick organic soils, International Journal of Wildland

596 Fire, 20, 418-429, 2011.

597 Boelter, D. H.: Hydraulic conductivity of peats, Soil Science, 100, 227-231, 1965.

598 Branham, J. E., and Strack, M.: Saturated hydraulic conductivity in Sphagnum‐

599 dominated peatlands: do microforms matter?, Hydrological Processes, 28, 4352-4362, 2014.

600 Brown, S. M., Petrone, R. M., Mendoza, C., and Devito, K. J.: Surface vegetation

601 controls on evapotranspiration from a sub‐humid Western Boreal Plain wetland,

602 Hydrological Processes, 24, 1072-1085, 10.1002/hyp.7569, 2010.

603 Devito, K. J., Mendoza, C. A., and Qualizza, C.: Conceptualizing water movement in

604 the Boreal Plains. Implications for watershed reconstruction. Synthesis report prepared for

605 the Canadian Oil Sands Network for Research and Development, Environmental and

606 Reclamation Research Group, 164, 2012.

607 Dixon, S. J., Kettridge, N., Moore, P. A., Devito, K. J., Tilak, A. S., Petrone, R. M.,

608 Mendoza, C. A., and Waddington, J. M.: Peat depth as a control on moss water availability

609 under evaporative stress, Hydrological Processes, 31, 4107-4121, 10.1002/hyp.11307, 2017.

610 Flannigan, M. D., Logan, K. A., Amiro, B. D., Skinner, W. R., and Stocks, B.: Future

611 area burned in Canada, Climatic change, 72, 1-16, 2005.





Frolking, S., and Roulet, N. T.: Holocene radiative forcing impact of northern peatland
carbon accumulation and methane emissions, Global Change Biology, 13, 1079-1088, 2007.
Gorham, E.: Northern peatlands: role in the carbon cycle and probable responses to
climatic warming, Ecological applications, 1, 182-195, 1991.
Hogan, J. M., Van der Kamp, G., Barbour, S. L., and Schmidt, R.: Field methods for
measuring hydraulic properties of peat deposits, Hydrological processes, 20, 3635-3649,

618     2006.

Hokanson, K. J., Lukenbach, M. C., Devito, K. J., Kettridge, N., Petrone, R. M., and
Waddington, J. M.: Groundwater connectivity controls peat burn severity in the boreal
plains, Ecohydrology, 9, 574-584, 10.1002/eco.1657, 2016.
Kasischke, E. S., and Turetsky, M. R.: Recent changes in the fire regime across the
North American boreal region—spatial and temporal patterns of burning across Canada and
Alaska, Geophysical research letters, 33, 2006.
Kennedy, G. W., and Price, J. S.: A conceptual model of volume-change controls on the
hydrology of cutover peats, Journal of Hydrology, 302, 13-27, 2005.
Kettridge, N., Tilak, A. S., Devito, K. J., Petrone, R. M., Mendoza, C. A., and
Waddington, J. M.: Moss and peat hydraulic properties are optimized to maximize peatland
water use efficiency, Ecohydrology, 2015a.
Kettridge, N., Turetsky, M. R., Sherwood, J. H., Thompson, D. K., Miller, C. A.,
Benscoter, B. W., Flannigan, M. D., Wotton, B. M., and Waddington, J. M.: Moderate drop in
water table increases peatland vulnerability to post-fire regime shift, Scientific reports, 5,
2015b.
Lewis, C., Albertson, J., Xu, X., and Kiely, G.: Spatial variability of hydraulic
conductivity and bulk density along a blanket peatland hillslope, Hydrological Processes,
26, 1527-1537, 10.1002/hyp.8252, 2012.
Lukenbach, M. C., Devito, K. J., Kettridge, N., Petrone, R. M., and Waddington, J. M.:
Hydrogeological controls on post-fire moss recovery in peatlands, Journal of Hydrology,
530, 405-418, http://dx.doi.org/10.1016/j.jhydrol.2015.09.075, 2015a.
Lukenbach, M. C., Hokanson, K. J., Moore, P. A., Devito, K. J., Kettridge, N.,
Thompson, D. K., Wotton, B. M., Petrone, R. M., and Waddington, J. M.: Hydrological





controls on deep burning in a northern forested peatland, Hydrological Processes, 29, 4114-
4124, 10.1002/hyp.10440, 2015b.
Lukenbach, M. C., Devito, K. J., Kettridge, N., Petrone, R. M., and Waddington, J. M.:
Burn severity alters peatland moss water availability: Implications for post-fire recovery,
Ecohydrology, 9, 341-353, 2016.
McCarter, C. P. R., and Price, J. S.: Ecohydrology of Sphagnum moss hummocks:
mechanisms of capitula water supply and simulated effects of evaporation, Ecohydrology, 7,

649    33-44, 2014.

McCauley, C. A., White, D. M., Lilly, M. R., and Nyman, D. M.: A comparison of
hydraulic conductivities, permeabilities and infiltration rates in frozen and unfrozen soils,
Cold Regions Science and Technology, 34, 117-125, http://dx.doi.org/10.1016/S0165-
232X(01)00064-7, 2002.
Petrone, R. M., Devito, K. J., Silins, U., Mendoza, C. A., Brown, S. C., Kaufman, S. C.,
and Price, J. S.: Transient peat properties in two pond-peatland complexes in the sub-humid
Western Boreal Plain, Canada, Mires & Peat, 3, 2008.
Prat-Guitart, N., Rein, G., Hadden, R. M., Belcher, C. M., and Yearsley, J. M.:
Propagation probability and spread rates of self-sustained smouldering fires under
controlled moisture content and bulk density conditions, International Journal of Wildland
Fire, 25, 456-465, https://doi.org/10.1071/WF15103, 2016.
Quinton, W. L., Hayashi, M., and Carey, S. K.: Peat hydraulic conductivity in cold
regions and its relation to pore size and geometry, Hydrological Processes, 22, 2829-2837,
10.1002/hyp.7027, 2008.
Roulet, N., Moore, T., Bubier, J., and Lafleur, P.: Northern fens: methane flux and
climatic change, Tellus B, 44, 100-105, 1992.
Sherwood, J. H., Kettridge, N., Thompson, D. K., Morris, P. J., Silins, U., and
Waddington, J. M.: Effect of drainage and wildfire on peat hydrophysical properties,
Hydrological Processes, 27, 1866-1874, 2013.
Shetler, G., Turetsky, M. R., Kane, E. S., and Kasischke, E. S.: Sphagnum mosses limit
total carbon consumption during fire in Alaskan black spruce forests, Canadian Journal of
Forest Research, 38, 2328-2336, 2008.





Šimůnek, J., Šejna, M., and Van Genuchten, M. T.: The HYDRUS-2D software package
for simulating the two-dimensional movement of water, heat, and multiple solutes in
variably-saturated media: version 2.0, US Salinity Laboratory, Agricultural Research Service,
US Department of Agriculture, 1999.
Smith, L. C., MacDonald, G. M., Velichko, A. A., Beilman, D. W., Borisova, O. K., Frey,
K. E., Kremenetski, K. V., and Sheng, Y.: Siberian peatlands a net carbon sink and global
methane source since the early Holocene, Science, 303, 353-356, 2004.
Stocks, B. J., Mason, J. A., Todd, J. B., Bosch, E. M., Wotton, B. M., Amiro, B. D.,
Flannigan, M. D., Hirsch, K. G., Logan, K. A., and Martell, D. L.: Large forest fires in Canada,
1959–1997, Journal of Geophysical Research: Atmospheres, 107, 2002.
Thompson, D. K., and Waddington, J. M.: Wildfire effects on vadose zone hydrology
in forested boreal peatland microforms, Journal of hydrology, 486, 48-56, 2013.
Thompson, D. K., Wotton, B. M., and Waddington, J. M.: Estimating the heat transfer
to an organic soil surface during crown fire, International Journal of Wildland Fire, 24, 120-

686     129, 2015.

Turetsky, M. R., Wieder, K., Halsey, L., and Vitt, D.: Current disturbance and the
diminishing peatland carbon sink, Geophysical Research Letters, 29, 2002.
Turetsky, M. R., Amiro, B. D., Bosch, E., and Bhatti, J. S.: Historical burn area in
western Canadian peatlands and its relationship to fire weather indices, Global
Biogeochemical Cycles, 18, 2004.
Turetsky, M. R., Kane, E. S., Harden, J. W., Ottmar, R. D., Manies, K. L., Hoy, E., and
Kasischke, E. S.: Recent acceleration of biomass burning and carbon losses in Alaskan forests
and peatlands, Nature Geoscience, 4, 27-31, 2011.
Van Genuchten, M. T.: A closed-form equation for predicting the hydraulic
conductivity of unsaturated soils, Soil science society of America journal, 44, 892-898, 1980.
Wieder, R. K., Scott, K. D., Kamminga, K., Vile, M. A., Vitt, D. H., Bone, T., Xu, B.,
Benscoter, B. W., and Bhatti, J. S.: Postfire carbon balance in boreal bogs of Alberta, Canada,
Global Change Biology, 15, 63-81, 2009.
Yu, Z., Loisel, J., Brosseau, D. P., Beilman, D. W., and Hunt, S. J.: Global peatland
dynamics since the Last Glacial Maximum, Geophysical Research Letters, 37, 2010.



Zoltai, S. C., Morrissey, L. A., Livingston, G. P., and Groot, W. J.: Effects of fires on
carbon cycling in North American boreal peatlands, Environmental Reviews, 6, 13-24,
10.1139/a98-002, 1998.