# Peer review of "Seasonally frozen soil modifies patterns of boreal peatland wildfire vulnerability"

_Hydrology and Earth System Sciences, 2017_

## Short Comment (SC1) · 26 Jan 2018

I read the paper by Dixon et al. entitled "Seasonally frozen soil modifies patterns of boreal peatland wildfire vulnerability" with considerable interest as it purports to couple frozen soil dynamics with fire vulnerability in boreal ecosystems. Without a doubt, this is an area requiring further understanding as the coupled heat-mass transfer in organic soils is vexing.

Unfortunately, I found this paper fundamentally flawed on a number of grounds. It has a large number of boundary and initialization conditions that do not make physical sense. The parametrization of frozen ground is unrealistic on many fronts. While I do not want to be overly negative, I present the authors a number of concerns and believe that this

type of paper would do more harm than good and be misleading to others as it is more an example of running Hydrus-2D than realistically simulating and understanding this system.

1) Hydrus-2D does not simulate freeze/thaw dynamics. There are several models that do. Having frozen ground simulated as static 'blocks' that do not move for up to 5 weeks is completely unrealistic, misleading, and ignores decades of research on heat and mass transfer in frozen soils. What the authors do is introduce and aquitard at some specified depth and keep it static for the period of simulation. There is no consideration of frozen ground physics, two-directional thaw, etc., despite dozens of papers on this regard. I think most people realize the role of aquitards in limiting water.

2) The paramterization and boundary conditions are incredibly unrealistic. How can we understand anything about this dynamic and coupled system when the appropriate thermal setup has not been accounted for? If in the spring the frost table was at the surface and allowed to descend, and the heat/mass transfer be accounted for, that would be of value and interesting results related to changing moisture conditions with thermal evolution (which would be heterogeneous because of variable soil thermal properties). There are models that do this computationally intense work.

3) The peat properties do not vary with depth. The authors cite one of their own studies to say this is true, but this ignores dozens of papers, and many in this environment that state otherwise. The Quinton et al. reference is misquoted and misinterpreted. Depth-dependent hydraulic and thermal properties are critical in understanding this process and the role of ice on moisture contents. This is ignored.

4) The upper boundary of 4.5 mm/d evaporation is unrealistically high. For 5 weeks? With no change in the geometry of frozen ground? At this point, I am unsure if this model has any grounding in all in reality the good fieldwork that has likely occurred in this area.

5) No rainfall for 50 days? This is unreasonable and likely imposed to 'prove a point'

with the model.

6) The rationale for setting conserving and productive peats below different hummocks/hollows is not justified.

7) The moisture recharge rate of 1.8 mm/d from the frozen soil layer is strange. Where is this moisture added to the domain? At the lower boundary? The upper boundary? At the aquitard. This is unclear and regardless incorrect without a moving boundary.

8) The 'continuity of frost layer being only important for productive peat' is an artefact of the model setup, not a real conclusion.

9) Is there any model validation?

10) Figure 5 and 6 looks like a screen cap and the quality should be improved.

In the end, I could comment on the interpretation of the results, but the results of this modelling exercise are not validated, the model setup is unrealistic, and all conclusions are dubious and speculative. I caution the authors to carefully consider these comments and whether this research is to 'prove' a clever point with an inappropriate model setup or truly trying to understand this system and its vulnerability to fire.

---

## Referee Comment (RC1) · Anonymous Referee #1 · 7 Feb 2018

As well articulated in the short comment on the manuscript by S. Carey, this paper is not suitable for publication in its present form.

First, the paper does address a relevant scientific question and is within the scope of HESS. The issue of frost in controlling soil moisture content in boreal landscapes (and the potential influence on fire) is an interesting topic. However, the model used (the well-established and regarded Hydrus 2D) simply isn't the tool for the job. The authors gloss over much of the details regarding model setup, ignore reality by presenting unrealistic boundary conditions, and then deliver bold and over-reaching statements as to the impacts.

Hydrus has been used to simulate permafrost and frozen ground before, but the literature has advanced well beyond this. Works by Evans, Ge, McKenzie, Kurylyk and

others all clearly show the advances and importance of appropriately simulating heat and mass in frozen ground (with or without permafrost). The simple geometry utilized to represent frozen ground along with the 'static' nature of frost (in an unrealistic climate boundary), the addition of water at some point somewhere, etc., simply provides model outputs that have no grounding in physical reality. It would have been much more informative if an appropriate model was compared with field data in a number of different scenarios. This would then allow simulations and 'gaming' regarding the role of ice. Clearly the descent and decline of the ice lenses is what is important here. Regarding the water conserving/productive peats, this concept is not well realized and I'm unsure of the authors rationale for its setup in the model is. Is this based on actual field data?

The conclusions of the model are speculative and not valid at this point. I suggest the authors take a more guided and nuanced approach by coupling an appropriate model with the undoubted wealth of field data in this region. From this, hopefully real insight into the role of frost position, descent and geometry have on soil moisture in near-surface soils will be realized.

---

## Referee Comment (RC2) · C. Maule (Referee) · 17 Feb 2018

Review; Seasonally frozen soil modifies patterns of boreal peatland wildfire vulnerability. Abstract: Submitted Interactive comment: GENERAL COMMENTS The discussion paper considers the role of evaporative drying of peatlands with an underlying shallow frost layer. The focus of drying is upon surface peat soils becoming dry enough that they can reach a smouldering threshold moisture content. The paper explores this using modeling of planar and hummocky peatlands. The subject matter is of interest and does have value with understanding hydrology of any frozen soil system. That of just the frost layers potential role in limiting moisture transfer from deeper parts of the soil and resulting in drier conditions is of interest (unless it has been explored by others for organic soils). Although this is a modeling approach it can help explain and

further direct interpretation of field data. The paper does require changes in organization, method explanation, and technical corrections to properly present and understand their findings. Overall I view these changes as major as it is difficult to assess the paper's outcome at this stage. To improve the paper the following need to be considered: use of terms needs to be standardized; assumptions need to be clearly stated at the beginning; the methods section describing models needs to be reorganized and information added; why a flat frost layer is used beneath hummocky terrain rather than a layer that follows surface contours; why both volumetric and gravimetric water contents are used; why it is necessary to use 'probability' instead of just water content; and addressing possible issues to do with numerical boundary errors between PR and WC domains.

Title is a bit misleading as the focus of the paper is upon how frost layers affect surface moisture contents. Low surface moisture contents in turn affect smouldering vulnerability. It is suggested that somehow the title have more a focus upon moisture contents.

Description of the model is scattered throughout first part of paper making it difficult to follow and understand. Some of this scatter was made necessary by having to first define terms, especially 'water conserving' and 'productive'. It would be more clear for the reader if the hydraulic properties (2.3.4 lines 191- 231) were first given then the model description (dimensions (section 2.3.2 Lines 133 to 154) and design (2.3.5, lines 234-248) were combined), then followed by model boundary and initial conditions (2.3.3, lines 155-231). Also use of terms describing model, especially regular model is confusing and it is suggested that the term 'planar' be used replacing 'regular' and 'topography excluded'. It is also not clearly understood what the initial moisture contents of the models are. It appears that the peats are initially saturated throughout – this is critical for the reader to understand – and that 'water table', as used by the authors, is synonymous with 'saturation'.

Description of where frost layer is located and type of frost layers tested is entirely missing in the methods section. Rational for using a 'planar' frost table (shown in Fig

4) rather than a frost table that is of constant depth beneath the ground surface must be given.

364-366 and 386-392: not clear how lateral water transfer results in zones of low GWC at the example A interface, which is the boundary between the PR and WC peat types. If water transfer was occurring then water would move from low soil water suction (high water content) to high soil water suction (low water content) due to an hydraulic gradient. A vertical 'tongue' of very low GWC extending downward in the WC peat type would not be expected. The authors must discuss that this is not an artifact caused by numerical calculations at model boundary conditions between the PR and WC boundary. If the density of peat is constant with depth and type then the hydraulic gradient has to have flow going from 'wet' to 'dry' thus flow could not occur from the WC side of the hummock to the productive hollow.

Assumptions and information on evaporation stated in lines 463 to 482 is informative. Some of the assumptions should have been clearly stated within the methods section

MORE SPECIFIC COMMENTS Initial confusion with effect of frost layer. At first it is implied to the reader that the frost layer can limit upward capillary flow from the water table (Line 66), but later it appears that it might also include the recharge of soil water to the evaporatranspiration needs of the layer above as the frost layer itself thaws and frees up water (Lines 175-179). In discussion of the results it is apparent that the frost layer plays both roles. These roles should be clearly stated within the first part of the paper.

Frost layer relative to water table and peat moisture content. It is not clear what the moisture content of the frost layer is. It appears that it forms within the water table and thus would be saturated. This needs to be clearly stated within Methods. If it is saturated then one can assume that it is 'relatively impermeable'.

Surface water tension. The use of this term is confusing, however if this term is preferred by the journal then ignore this comment. First 'tension' implies water potential

relative to atmospheric and thus in the context of unsaturated soils 'tension' should be accompanied by negative values (e.g -400 mb). The second source of confusion is the use of 'surface' which can be misleading with 'ground surface', 'evaporating surface' or 'water table surface'. It is recommended that the authors use the term 'soil water suction' in its place as 'suction' has the negative term within the word. This will also avoid further confusion for such instances as line 335 "near-surface water tensions'

Use of VWC, VMC, and $\theta$ to represent volumetric water content. It is suggested that where possible only $\theta$ be used, however as this is a symbol and for sentence structure purposes VWC could be used.

A definition of 'smouldering' would help understanding of article by readers not familiar with peatlands (lines 41-51). I had to look it up to understand the issues see Rein et al 2009: "Smouldering is a flameless form of combustion, deriving its heat from heterogeneous reactions occurring on the surface of a solid fuel when heated in the presence of oxygen [3]. The fundamental difference between smouldering and flaming combustion is that in smouldering, oxidation of the reactant species occurs on the surface of the solid rather than in the gas phase. The characteristic temperature, spread rate and heat released during smouldering are low compared to those in the flaming combustion of a solid. Smouldering fires in forest biomass propagate on average at around of 10-30 mmÂůh-1 and the peak temperature is around 550-650 $\circ$C [4]." (p2).

Smouldering threshold. A threshold GWC of 250% is given (line 263). As this is a key value to discussion a reference or two supporting this threshold would strengthen its use.

GWC as a % or a ratio. As GWC is defined by equation 5 (252) as a ratio its use throughout the paper should be as a ratio and '%' values avoided.

Density value used for peat in models. As both VWC and GWC are used interchangeably throughout the paper it would help the reader if the dry bulk density of peat was given. Clarify whether one value was used for all models and depths or several or a

**HESSD**

range. For equation 5 (252) clarify that this is the dry bulk density used. Also correct conversion should also include the density of water in the equation and in the calculations – especially if kg m-3 are being used as units.

Use of terms to describe the 'regular' or 'planar' peatlands. This changes throughout the paper and makes it very confusing to follow. Objective ii) (lines 87-89) does not state the type of peatland topography however throughout the rest of the paper refers to objective ii) as being the 'regular domain' (134, 156, 237), a 'planar surface model' (157), and as 'topography excluded' (294). It is suggested that 'planar model' be used as 'regular' and 'topography excluded' are not descriptive enough. This term should also be used in objective ii).

Use of term 'microtopography' to represent hummocky peat terrain. This is less confusing however as it represents objective iii) (89) it needs to be clearly stated in objective iii) and defined so that 'peatland microforms' is understood by the reader to represent 'microtopography', or hummocky terrain variations of 0.4 m in amplitude and 2 m in wavelength. If 'microtopography' is to be the term of choice then it should be used consistently throughout and not other terms (e.g., 'topography included' line 348).

Consistency with use of other terms throughout paper; e.g., 'seasonal ice dynamics (70)', 'seasonal frost layers' (76), 'near-surface frozen soil thaw dynamics' (79). 'depth to ice' meaning 'depth to frozen soil' (86) 105: 'no ice'. Is 'ice layer' different from 'frost layer'?

139: soils tested in McCauley et al 2002 were mineral soils (highest om content for one was 9.5%) and hydraulic conductivity was measured with liquid fuel. A reference with peat soils would be more appropriate or mention the limitations of this reference. Of interest with McCauley et al (2002) is that K was measured at different frozen moisture contents thus showing K was a function of degree of ice-saturation. If the frost layer of the models was saturated or near-saturation then permeability can be assumed to be very low.

[Figure]

Does using the metal probe to indicate frost layer only work if the peats are above a certain moisture content?

140-142: clearly state that 'rectangular geometric objects' and 'inactive objects' refer to the frozen layer. This is confusing as the terms 'layer', 'cells', and 'objects' are used to mean the same thing

143, 146, 158: upon first use define evaporating surface (rather than at line 169) and locate in z plane. Does it always coincide with the ground surface? If so then perhaps state this

158: for the starting water table depth would the evaporating surface always coincide with the ground surface?

188: by 'planar' water table depth within the microtopography model this means it is 'flat' and given a depth of 0.04 m from base of hollows it would thus be 0.44 m deep from hummock tops? If so then add phrase "..0.04 m from base of the hollows and 0.44 m from hummock peak and higher…"

Consider showing a figure (with dimensions) that show the planar and microtopographic domains with labelling demarcating water conserving hummocks and hollows from productive hummock and hollows and the different frozen soil geometries. This would help readers better visualize the models.

204: try to improve upon definitions or terms used to describe WC and PR. WC: 'peat profile displayed high surface tensions under evaporation' high surface tensions = high suctions meaning the peat dried out??. PR: 'able to maintain low surface tensions during evaporative stress' meaning low suctions and the peat remained wet?? I did find the terms 'water conserving' and 'productive' not intuitively representing what these peats. Why 'water conserving' if the peat dries out? Why 'productive' if the peat conducts water and remains wet?

Do not understand the need to show probability of ignition vs time (eg fig 5) and Mean

[Figure]

GWC vs time figures when they are both the same (shapes are the same)?

TECHNICAL DETAILS

45-46: Prat-Guitart et al., 2016 should be cited last in the list citations to keep with chronological order established in rest of article

131-132: Šimůnek et al., 1999?

154: space between 1.4 and 'm'. Check throughout paper as there are other instances.

167: add a reference for 'Environment Canada, 2017'

199: citations as part of the sentence (as opposed to within brackets) are now italicized where as the few before (e.g. lines 101, 121) were not. Also square brackets [ ] are now used rather than rounded ( ). Check throughout paper

203: Kettridge et al. missing a period and the 2015 is missing an 'a' or a 'b'.

216: Quinton et al missing a period

219-221: explicit definition of 'water conserving' and 'productive' should be before use in lines 204-205

231: perhaps table should clarify what the models are with each material; it is not clear if 'mean', 'water conserving', and 'productive' are associated with the planar model or not?

253: specify whether  refers to particle density or bulk density (and if bulk density whether it is wet or dry). By unit analysis it will have to be dry bulk density. Also specify that $\theta$ is VWC. To make Eq 5 correct with regards to dimensions the density of water term must be included: GWC = $\theta$ w/b, where b is dry bulk density

261: clarify symbols in Eq 6 and 7 by providing definitions in text of $\mu$p (mean peat density?) and $\sigma$p (standard deviation of peat densities)

263: should use GWC as a ratio throughout paper not a percentage. This gets confus-

ing for use of 'x' in equation 7.

265: is x expressed as GWC in kg/kg?

281: to avoid confusion it should be stated that 'moisture recharge' is referring to moisture released from the ice as it melts (lines 175-177) as opposed to other sources of recharge; i.e. rainfall (line 187) or capillary rise.

299: as figure uses hours then use hours in text to help reader; e.g., "after 72 hours (3 days). . ."

300: "..after 168 hours (one week).."

306: "day 7 (hour 168).."

299: 307: Figure 2: difficult to visually differentiate 'productive' from 'mean' lines at 100% screen magnification.

308: The 0.20 m depth refers to top of the frost layer? Why is this different from 0.15 m depth used for the planar model (line 141)?

314: as figure is in hours suggest the following phrasing 'After three weeks (504 hours) of . . ."

308-309. Is this figure just for the microtopographic model? Or does 'mean' refer to the planar model? Clearly state within Figure title and within text discussing figure.

316-332: not clear whether this section is this referring to planar or microtopographic models.

322, 325, 328 what is meant by Figure 3b/c, 3bc? Figure 3b,c?

324: as $\theta$ is VWC just state "..a $\theta \approx$ 0.50 m3 m-3.

342: Figure 3. State whether the figure is for planar or microtopograhic model. As model dimensions are 5 m why is 4 m used here?

342: Figure 3. Shading used makes it very difficult to discern $\theta$ values between 0.3 and 0.5. Could use 3 colors in shading (as done with Fig. 4) rather than just 2. Why is Fig 3 as volumetric whereas the rest of the paper and figures discuss gravimetric?

351-352: remove the 'and' in the sentence to improve sentence structure.

353: Figure 4. Gravimetric color bar in figure should be expressed as a ratio not a %

353: Figure 4. Ice layer depths and configurations are not described in the text. Within figure it is apparent that they are all planar regardless of surface topography – needs to be discussed why this was assumed as opposed to having a set depth beneath the surface, which would be more reflective of field conditions.

353-362: to avoid confusion change the top titles of the figures from 'after 2 weeks' to '2 weeks' and '5 weeks'. Then for line 358 remove the 'after' so it reads "scenarios at two weeks. For the results at five weeks,...". Otherwise the reader is considering a time period following 2 or 5 weeks.

367: as there are no figure sublabels within Figure 4 it is difficult to discern what 'Figure 4a/b' refers to.

383: Figure 4 does not show ice scenarios at three weeks.

394: use the same terminology throughout the paper for 'discontinuous' frost layer. Figure 4 used 'discontinuous' thus Fig 5 should not use 'blocks' 395: figure is titled 'probability of ignition' which refers to equation 8 (line 269). This is misleading and that of what the equation represents 'probability of the gravimetric water content being lower than the threshold for smouldering' (lines 262-266) would be more accurate.

402: might have missed this before but state what peat density (or densities) were used in the model – or at least state if only one peat density (by depth and peat type) was used or a range.

424-431: specify that this is for the planar model. This series sentences are confusing

as it starts by stating for the 'prescence or absence of a frost layer', however fig 2 is just for a solid layer frozen conditions and peat DID NOT remain saturated over multiple weeks of evaporation for the productive domain (difficult to tell the lines apart)

431: do you mean Figure 1?

435: sentence is awkwardly phrased

463-473: some of this information; fixed evaporation rate of 4.5 mm/d should be stated in methods.

540: '2015a' should be cited before '2016'

627: volume and page numbers needed for Kettridge et al., 2015a and 2015b. Kettridge et al 2015a was published in Ecohydrology in 2016.

---

## Referee Comment (RC3) · Anonymous Referee #3 · 27 Feb 2018

The paper is a modelling study to investigate moisture dynamics during the thawing period in the northern peatland. The scientific question of the paper is pressing for cold region hydrology and relevant for the HESS. However the methods used in the paper are not appropriate to get results and make conclusions suitable for publication in HESS.

The model and model setup have several important limitations:

1) The model doesn't account for heat transfer in the soil profile but it is used for investigation of frozen layer influence on water transfer.

2) Frozen layer is assumed to be impermeable and permanent in time at the same depth for several weeks that is not a case in natural conditions where thawing/freezing

front is constantly moving

3) Model outputs are not compared with any observed soil water content data to evaluate the model performance

4) The model does not take into account changing weather conditions at the peat surface like air temperature, air moisture and rain

5) Fixed daily evaporation rate of 4.5 mm/day looks unrealistic

6) Initial conditions are set in arbitrary way. The soil just after the snowmelt is not necessarily thawed and fully saturated. It could be frozen with different degree of saturation depending on autumn weather conditions

7) Statement "all these assumptions (no thaw, evaporation rate, hritA and initial conditions) are within 1-2 mm/day, and are constant between scenarios" (lines 481-482) looks unfounded in terms of quantitative assessment.

The chosen simulation design does not reflect dominant natural processes that govern soil moisture and water table dynamics: heat transfer and water phase change within the profile, variable in time air temperature, air moisture and precipitation and moving thawing/freezing front. It is not validated with observations and thus could not be used as a ground for drawing conclusions about water behavior in real peat profile under natural conditions.

---

## Author Comment (AC1) · 9 Mar 2018

We thank the reviewer for their interest in this work and for their comment. We have addressed their concerns as to the modelling set up in detail below. In summary, many of the concerns they raise are do not map onto the specific aims of this study listed at the end of the introduction. This study is a reduced complexity, exploratory modelling exercise. The purpose of which is to evaluate how hydraulic properties and frost layer continuity (through their role as aquitards) interact to drive vertical and lateral transfers and induced heterogeneity in patterns of vulnerability to smouldering during prolonged rain free periods. We further reiterate this targeted aim in the methodology section (line 174) and discussion (line 484), highlighting that the modelling is only able to give insights into behaviour, not predictions.

[Figure]

We acknowledge that the reviewer has not recognised the aims of the study. Therefore we aim to minimise such a misunderstanding by emphasising our aims within the abstract, and adding a short section at the end of the introduction section (after the 3 aims are stated). In this added section, we will clearly state the areas which are beyond the scope of the study. We will also thoroughly check the manuscript to confirm all language is used clearly and directly map to these aims. A key point is that this study does not attempt to simulate or recreate detailed thermal and water balance calculations for a specific peatland, under specific measured conditions. This study is exploratory (or heuristic) modelling in which a simplified modelling framework and boundary conditions are used within in a verified numerical model to generate insights into the directionality and relative magnitude of system response to forcings.

To recap the modelling aims:

* explore how hydraulic properties and frost layer continuity interact to drive vertical and lateral transfers of water during prolonged rain free periods of high fire risk.

* explore how spatial variability in hydraulic properties, peatland microforms and frost layers interact to induce spatial variability in GWC and associated smouldering severity during the exceptionally dry periods which precede wildfires.

Here, we impose clear and tight limits on what we wish to explore. We examine two degrees of freedom – peat properties and frost layer (aquitard) continuity and only explore how changing these alters the degree of vertical and lateral water flow between scenarios. In the second aim above, we also introduce peatland microtopography. We are therefore only comparing relative changes in behaviour. Furthermore, we restrict our modelling to very dry periods.

Additional text to be added after the aims above (at line 91 in original manuscript):

"It is important to emphasise that the numerical modelling in aims ii and iii uses a heuristic modelling framework, in which a simplified representation of the system is

used to explore relative differences between modelling scenarios (Bankes, 1993). The purpose is to illuminate the role of different controls on the study system, and not to provide explicit quantitative predictions. In this respect, the results of the modelling should be treated in a comparative sense."

Bankes, S 1993 Exploratory Modeling for Policy Analysis. Operations Research 41 435-49.

In numerical modelling a model set-up can be both verified (which means it has been established that the model accurately represents the processes it is designed to, i.e. the equations are faithfully represented), and validated (which means the set-up of the model has been compared to known input and output data to establish it can accurately recreate an observed event). In this study the verification of the model is accomplished by use of a widely used numerical model/software package for simulating water movement in the unsaturated zone. But the model is not validated, as the purpose is not to recreate or simulate a known event and then extrapolate from it. Many of the points the reviewer raises are relevant to a modelling exercise which aims to simulate, or recreate detailed water balance calculations for a specific peatland, under specific, measured conditions, i.e. to validation of the model. See also point 9 below where we link to a paper in Science which discusses the appropriateness of model validation and verification.

We provide comments to the reviewer's numbered points below:

1) The parameters and boundary conditions that the reviewer has mentioned are stated in the methods, so we are clear and up front about this limitation throughout. In the case of the frost layer, and the lack of dynamics, this is inherently a simplification of the system. However, the aims of the study states that we are only looking to explore the effects of the frost layer on vertical and lateral transfer of water during extreme drying events. While the role of aquitards is well understood, interactions between a seasonal frost layer and different peat hydraulic properties have not been demonstrated. This

issue cuts to the heart of how complex a model needs to be in order to generate new insights. To date, simplified one and two-dimensional models have provided key insights in the fields of hydrology and ecohydrology. Indeed, Benscoter et al, 2008 show with a simple peat smouldering model that the most important variables are moisture content and peat properties. Even though a model does not directly represent all of the complexity in an uncertain and unknown system, it is still able to teach us something about the system. In the discussion section, we address the recharge effect from melting ice (and other assumptions) at line 463 and show it is of the same order of magnitude as other assumptions. Most importantly, our results can very broadly be divided into those for which the water table recession reaches the aquitard, and those it does not. In very simple terms it is only those simulations in which the water table drops to the depth of the aquitard that would be markedly influenced by its recession and/or melt recharge. In these cases, the overall relative behaviour of the system in this reduced complexity model wouldn't appreciably change with ice recession dynamics (the drier simulations would still be drier than the wetter ones), but timing when the system reaches specific states would change – taking longer to reach a specific point in the event of melt water recharge. This is discussed at line 484

2) Although we agree with the reviewer that questions of heat/mass balance in peatlands are interesting and would be worthy of future study, that is not within the aims or scope of this study. In the context of this study, we highlight that the thawing of ice from the frost table represents a small change in store. Relative to evapotranspiration occurring, this is only a maximum of 1.8 mm/day assuming the soil was frozen whilst saturated. In this study we are not trying to replicate the smouldering process, or conduct detailed water balance simulations.

3) We parameterise peat properties based on a summary from one of our own papers. However, this paper also contains data from extensive field data collection from other studies, (at a site where the modelling is based) and utilizes peat properties from field samples at other sites (n~300). Although these results show weak depth dependence

with peat hydraulic properties over the top ∼60cm, depth explains only a small proportion of variance, we have attached a basic sketch plot of alpha vs depth for hummock samples using the data from the aforementioned paper for example (alpha on the X axis, depth in cm on the Y axis).

Furthermore, the relationship between depth and properties is not unidirectional; in some cases the average properties (for example alpha) of a depth layer across all samples do not linearly decrease but fluctuate up and down as one moves down the "average" depth core. Given the substantial uncertainty in the precise numeric relationship between depth and peat properties, the use of uniform values for different microtopographic features, reflecting the range of variability in peat properties, is a justifiable simplification. As an aside, more extensive, critical inter and intra site investigation into the relationship between depth and peat properties in the near surface represents an exciting area for future study, both in the field and in modelling.

We will reword the Quinton reference, as we believe the reviewer has read that the whole sentence refers to the reference. Rather, the first part before the comma is intended to talk about the Quinton study, while the subsequent clause places their findings into the context of our argument. We will reword for clarity as follows:

Quinton et al [2008] also showed that Ks is dependent on the degree of compaction and decomposition; however, compaction and decomposition does not necessarily show a linear relationship with depth.

4) The section detailing evaporation is poorly worded, as the rate is for PET not AET, (which will be lower than 4.5mm/day), and we propose amending the start to read:

"We assume a fixed daily Potential Evaporation Rate (PET) of 4.5mm/day; this is a fairly high. . .."

As stated in the aims, the purpose of the modelling is to explore systems behaviour during exceptionally dry, rain-free periods. As stated a line 470: "..this is a fairly high rate,

however, Sphagnum evapotranspiration rates can reach an average of 0.47 mm/hr, and unlike systems which show highest ET rates coincident with highest air temperatures, Alberta peatlands show peak ET rates during the early growing season as frozen soil is thawing (Brown et al., 2010), which is the period being modelled". This means that the AET.

5) As stated in the aims, we are modelling an exceptional rain-free period. Furthermore, we do not report and/or discuss results solely from the perspective of the system after 50 days. We look at how the system response changes throughout the drying period; the bulk of our discussions focus on 2 weeks and 5 weeks (35 days) from the start of the model simulations. It is important to emphasise that the results at 2, 3, 5 weeks, etc., will be the same regardless of whether the total model run time is 5 weeks, 50 days, or even longer. Lastly, it is important to reiterate again, this is not intended to be a simulation of a measured event. The purpose is to explore how the system responds, both in the presence and absence of the frost layer, to a sustained drying event.

6) We will included a brief explanation in the methods of the reasons for different peat properties. Currently reads (line 244)

"..the model domain was parameterised as; a water conserving hummock, a water conserving hollow, a productive hummock, a productive hollow, and a water conserving hummock. The sequence ensures all four possible transitions between hummock and hollow properties are represented in the model domain."

We will add following this:

"These peat types are based on the work in Dixon et al, 2017 and Kettridge et al, 2015; these previous studies showed that despite a wide range of possible peat hydraulic properties water balance responses to evaporative stress are bimodal. Peat tends to either be able to maintain low near surface tensions under evaporative stress and thus is able to remain productive and sustain evaporation through vertical movement

of water from deeper in the peat profile, or alternatively, peat will rapidly experience high near surface tensions, restricting evaporation and conserving water. This bimodal response to evaporative stress is observed in both hummock and hollow microforms. By alternating the peat types along a microtopographical transect, the model will allow us to explore how peat microforms with different representative hydraulic property types interact to alter local water balance at their interfaces"

7) The water recharge rate of 1.8 mm/d is calculated from the field work and explained at line 280-281, it is important to emphasise this is a maximum possible rate, based on the assumption that the peat is frozen in saturated conditions; the actual recharge rate may well be lower. We have then restated the maximum recharge figure in the discussion of model limitations/assumptions at line 469. However, we believe the way this is stated is not clear enough and has led to the reviewer's confusion. We will change to:

"We also assume the frost layer does not thaw during the model runs and thus does not provide water recharge in the model; for context, from the above field results such a recharge would be a maximum of approximately 1.8 mm/day, depending if the peat was saturated prior to freezing."

8) In exploratory (and indeed computational/contributive ) numerical modelling all results and conclusions are directly derived from the model set-up decisions. The peat properties are parameterised from field measurements, and based on other studies are divided into representative "types" which represent the two broad types of responses to evaporative stress in peatlands. Naturally, as one type of peat is able to maintain evaporation whilst minimising tensions, whilst the other type cannot, the first type loses more water to evaporation. So whilst we have chosen to represent these two types of peat in our modelling, and from that respect the model set up informs our results, this peat parameterisation is grounded in the results and conclusions of other studies. It is also important to emphasise that the gaps in the ice lens/aquitard are below both a productive hollow and a water conserving hollow, so where we find the productive

hollow is sensitive to the continuity of the ice lens, this is in comparison to the water conserving hollow. The key is that the continuity of the lens, in terms of effects on water flow, is most important in this location/this combination, in comparison to the other combinations/locations we have modelled.

9) No model validation is performed, as this is an exploratory, not a predictive piece of modelling. With an exploratory modelling approach, concepts such as model validation and sensitivity analysis can be seen as nonsequiters (see Bankes, 1993). Because the initial model set and the scenarios used in exploratory modelling are not simulations of specific times and places there is nothing to validate model output to. This also relates to general concepts of verification and validation as discussed in the hydrology literature (and more widely – see Oreskes, N, Shrader-Frechette, K & Belitz, K 1994 Verification, validation, and confirmation of numerical models in the earth sciences. Science 263 641-46. for example). In the respect of exploratory hydrological modelling the key principle is that the model used is verified, that is to say there is confidence that the model is mathematically representing the processes it is designed to do (which is the case with Hydrus as an established hydrological model)

10) The in-line figures for the review are .png files imbedded within the pdf document as requested by the journal. The final figures for journal production we have as separate .TIF files at 900dpi. However, should a further review copy of the paper be needed we will endeavour to improve the resolution of the imbedded .png files for the reviewers/editors.

We thank the reviewer again for taking the time to comment on the manuscript, and to help us try and improve it for publication. We trust that our more extensive descriptions of exploratory modelling concepts has put their mind at rest that we are not trying to quantitatively simulate or provide predictions (nor have we claimed to). It is also important to emphasise that this work is not, and does not claim to be, providing definitive answers. Rather, we hope to provide some insights into system behaviour that can then be critically explored in the field, and in more computationally complex predictive

numerical models. We feel that the ability of exploratory modelling to show that frost lenses can be important to seasonal water balance, but that this is not universal, and depends heavily on peat hydraulic properties is a useful insight, which advances our understanding of system behaviour and can inform future, more complex, hydrological water and mass balance modelling.

―――――――――――――――

[Figure]

[Figure]

Fig. 1.

---

## Author Comment (AC2) · 9 Mar 2018

We'd like to thank the reviewer for reading the paper and for providing comments to help improve the manuscript. We agree that the issue of understanding controls on near surface water balance and the links with seasonal frost and wildfire potential is a relevant question that is deserving of attention.

The reviewer has referred back to the detailed comments provided by S. Carey and so we will not duplicate some of our replies here, but restrict ourselves to addressing their specific comments.

The reviewer highlights that the Hydrus 2D model is unsuitable for replicating the system in question, as it is not capable of simulating heat and mass transfer in frozen

ground, and uses only simplified geometry for the frozen layer(s). On this point we completely agree with them, however, the aims and objectives of this study are not to simulate all the processes within the target system. As detailed in the response to S.Carey, the study is using exploratory (or heuristic) modelling in order to understand the magnitude and directionality in water balance response to the presence of a frost layer acting as an aquitard. This can be considered as a first step in understanding how near surface water balance can respond to the presence of a frost layer acting as a barrier to vertical flow of water from deeper in the soil profile. By isolating one aspect of the system (ice layer as barrier to water flow) in a simplified numerical modelling framework we are able to explore the comparative influence of this effect on near surface water balance. The reviewer comments that "clearly the descent and decline of the ice lenses is what is important here", however, without first establishing that the presence of the ice lenses can affect the near surface water balance (as we show it can only under some conditions here) the dynamics of this feature would be a moot point. Indeed, as we show in the manuscript there are peat types which will not lose substantial amounts of water through evaporation during prolonging dry periods, and for these peat types the decent and decline of the ice layer will have no effect on their near surface water balance. This is the inherent value and interest in using simplified numerical modelling set-ups - the ability to isolate different variables and examine the degree to which they influence the overall system behaviour.

As detailed in our response to S Carey, we propose to add clarification of the heuristic modelling aims at the start of the manuscript to help prevent confusion, and to expand the aims section to clearly state the areas which are beyond the scope of this exploratory study.

The reviewer has some concerns regarding the water conserving/productive peat properties concept, however, as detailed in the methods section this is based on previously published work. In response to S.Carey's comments we have suggested some amendments to provide an additional precis of the concept for readers who have not read the

other papers, which will further clarify the rationale in using these values.

The reviewer also expresses concerns that the conclusions are speculative and not valid, however, within the context of an exploratory modelling framework the conclusions are supported by the modelling and are restricted to comparative conclusions between our modelling scenarios. We agree that more detailed and complex numerical modelling could provide insights into the role of frost decent and geometry on near surface water balance, however such detailed predictions are beyond the narrower scope of this study.

---

## Author Comment (AC3) · 9 Mar 2018

We thank the reviewer for taking the time to read the paper and to provide comments on the manuscript to improve it, it is very much appreciated. We feel that their comments are helpful in identifying and improving areas of the manuscript, particularly in terms of clarifying some of our statements and in better explaining the aims and scope of the study to readers. We have addressed their specific numbered points below:

The model and model setup have several important limitations: 1) The model doesn't account for heat transfer in the soil profile but it is used for investigation of frozen layer influence on water transfer. 2) Frozen layer is assumed to be impermeable and permanent in time at the same depth for several weeks that is not a case in natural

[Figure]

conditions where thawing/freezingfront is constantly moving

Both of these first two points relate to a misinterpretation of the focused aims of the modelling study. The study is a reduced complexity, exploratory modelling study in which we are attempting to investigate the potential role of a seasonal frost layer on limiting vertical transfer of water from deeper in a soil profile to the evaporating surface, and whether this function has the potential to lead to enhanced drying at the near surface. We do not set out to (or claim to) simulate the actual water balance in a measured peatland. The limitations in the model set up and the potential effect of these limitations on the results are already discussed in the paper. However, in the response to S Carey we suggest that we need to make the scope of the study much clearer and to include a section after the aims where we explicitly state what is beyond the scope of the study, especially given that two readers have made the same comments assuming the study is predictive modelling. Furthermore, again also in response to S.Carey, we suggest an explicit statement in the abstract is needed regarding the exploratory modelling approach used so readers are aware right at the start of the paper, rather than it being introduced in the introduction and methods. In the response to S Carey we propose adding the following after the aims:

"It is important to emphasise that the numerical modelling in aims ii and iii uses a heuristic modelling framework, in which a simplified representation of the system is used to explore relative differences between modelling scenarios (Bankes, 1993). The purpose is to illuminate the role of different controls on the study system, and not to provide explicit quantitative predictions. In this respect, the results of the modelling should be treated in a comparative sense."

Bankes, S 1993 Exploratory Modeling for Policy Analysis. Operations Research 41 435-49.

3) Model outputs are not compared with any observed soil water content data to evaluate the model performance

[Figure]

This point is also raised, by S Carey and we copy our reply below: No model validation is performed, as this is an exploratory, not a predictive piece of modelling. With an exploratory modelling approach, concepts such as model validation and sensitivity analysis can be seen as nonsequiters (see Bankes, 1993). Because the initial model set and the scenarios used in exploratory modelling are not simulations of specific times and places there is nothing to validate model output to. This also relates to general concepts of verification and validation as discussed in the hydrology literature (and more widely – see Oreskes, N, Shrader-Frechette, K & Belitz, K 1994 Verification, validation, and confirmation of numerical models in the earth sciences. Science 263 641-46. for example). In the respect of exploratory hydrological modelling the key principle is that the model used is verified, that is to say there is confidence that the model is mathematically representing the processes it is designed to do (which is the case with Hydrus as an established hydrological model). In numerical modelling a model set-up can be both verified (which means it has been established that the model accurately represents the processes it is designed to, i.e. the equations are faithfully represented), and validated (which means the set-up of the model has been compared to known input and output data to establish it can accurately recreate an observed event). In this study the verification of the model is accomplished by use of a widely used numerical model/software package for simulating water movement in the unsaturated zone. But the model is not validated, as the purpose is not to recreate or simulate a known event and then extrapolate from it.

4) The model does not take into account changing weather conditions at the peat surface like air temperature, air moisture and rain

The purpose of the modelling exercise is to apply an evaporative forcing to the model over a prolonged period to look at the response of the system over time. Therefore, as this is not a simulation of a specific event or measured year we do not apply changing weather conditions, in a conceptual sense (as mentioned in the methods) the model runs can be thought of as representing a prolonged rain free period in Alberta, hence

there is no rainfall. The model does not directly include air temperature or relative humidity, but represents these through PET (which can be time variable). In the case of our model investigation, and in keeping with representing a conceptual period of prolonged drying, we apply a PET of 4.5mm/day on a diurnal cycle with hourly timesteps to represent the hourly change in PET during each daily cycle.

5) Fixed daily evaporation rate of 4.5 mm/day looks unrealistic

This is poorly worded and was also raised by S Carey, the figure is actually a fixed value for PET, not AET. We propose amending the start of the sentence in question to clarify this:

"We assume a fixed daily Potential Evaporation Rate (PET) of 4.5mm/day; this is a fairly high. . .."

6) Initial conditions are set in arbitrary way. The soil just after the snowmelt is not necessarily thawed and fully saturated. It could be frozen with different degree of saturation depending on autumn weather conditions

We agree with the reviewer that the rationale for initial water conditions are not well defined in the methods. We propose to add the following section at line 160, and then break the paragraph following this section, to start a new paragraph at "The base and sides. . ."

"It is important to note that our initial model set up has a relatively shallow starting water table. Conceptually, this represents near saturation of the peat following spring snow melt, which depending on the preceding conditions each winter/spring may not always occur. The purpose of this exploratory modelling investigation is to determine the relative magnitude of low near-surface water contents, given different scenarios for seasonal frost lenses. In this respect we choose to initiate model runs with comparatively high near-surface water contents, to ensure that any low modelled water contents over the model run time reflect the dominant processes leading to water loss from the

system, and not as a result of dry initial conditions applied in model set-up."

7) Statement "all these assumptions (no thaw, evaporation rate, hritA and initial conditions) are within 1-2 mm/day, and are constant between scenarios" (lines 481-482) looks unfounded in terms of quantitative assessment.

We agree that this section is poorly worded and somewhat glosses over an important point, which could benefit from a more in depth exploration in the text. We propose to greatly expand this discussion of assumptions in the context of exploratory modelling:

"There are therefore four broad input parameters within the model set up which will affect the resulting water balance; either by providing more or less water to the soil profile (no thaw mechanism, and initial water contents), or result in a greater or lesser volume of water being removed at the evaporating surface per time step (hcritA, PET). It is important to note that over the length of a model run these assumptions in model set up are all in the range of $\pm$1-2mm/day. Furthermore, as explored in the discussion, making different decisions with respect to these parameters would not dramatically change the overall system behaviour, but instead would change the timings that the system reaches different stages of near surface GWC. These assumptions are justified within an exploratory modelling framework, as the objective is to determine the trajectory of system behaviour and the relative magnitude of response between different scenarios, not to deliver quantitative predictions."

As with review comments from S Carey and review 1, we feel that we have not set out the exploratory nature of the modelling investigation early enough in the manuscript and this has led to misunderstandings about the scope/aims of the investigation. We trust that expanding upon our explanations of the exploratory modelling concept and by proposing to add in clear statements of what is beyond the scope of the study after the aims we can demonstrate that simplified numerical modelling can deliver important insights into peat hydraulic behaviour that the community can build upon in the future.

[Figure]

678, 2018.